# Can LLMs Reason Like Automated Theorem Provers for Rust Verification? VCoT-Bench: Evaluating via Verification Chain of Thought

**Zichen Xie** [1]   **Wenxi Wang** [1]

## Abstract

As Large Language Models (LLMs) increasingly assist secure software development, their ability to meet the rigorous demands of Rust program verification remains unclear. Existing evaluations treat Rust verification as a black box, assessing models only by binary pass or fail outcomes for proof hints. This obscures whether models can systematically reconstruct the explicit deductive steps required for verifying nontrivial Rust code. To bridge this gap, we introduce VCoT-Lift, a framework that lifts low-level solver reasoning into high-level, human-readable verification steps. By exposing solver-level reasoning as an explicit Verification Chain-of-Thought, VCoT-Lift provides a concrete ground truth for fine-grained evaluation. Leveraging VCoT-Lift, we introduce VCoT-Bench, a comprehensive benchmark of 1,988 VCoT completion tasks for rigorously evaluating LLMs' ability to reconstruct the entire verification process. VCoT-Bench measures performance along three orthogonal dimensions: robustness to varying degrees of missing proofs, competence across different proof types, and sensitivity to proof locations. Evaluation of ten state-of-the-art models reveals severe fragility, indicating that current LLMs fall well short of the reasoning capabilities exhibited by automated theorem provers.

## 1. Introduction

Rust (Matsakis & Klock, 2014) has emerged as the cornerstone of modern systems programming, adopted by the Linux kernel and major technology infrastructure for its rigorous guarantees of memory safety, concurrency correctness, and high performance. However, as the software industry increasingly relies on Large Language Models (LLMs) to

---
[1]University of Virginia. Correspondence to: Zichen Xie <graysonxie@virginia.edu>, Wenxi Wang <wenxiw@virginia.edu>.

*Proceedings of the $43^{rd}$ International Conference on Machine Learning*, Seoul, South Korea. PMLR 306, 2026. Copyright 2026 by the author(s).

synthesize complex code (Zhuo et al., 2024; Zhang et al., 2024; Ishibashi & Nishimura, 2024; Yang et al., 2025b; Zhang et al., 2023), subtle errors can be introduced to make the programs *functionally incorrect*. Consequently, formally verifying the correctness of Rust programs, by mathematically proving that programs satisfy precise specifications under all possible inputs, has become increasingly critical for establishing reliable and trustworthy systems.

Verus (Lattuada et al., 2024) is a state-of-the-art formal verification framework for Rust programs. A Verus program consists of three components: (1) executable Rust code, (2) specifications that precisely describe intended behavior through preconditions (`requires`) and postconditions (`ensures`), and (3) user-provided proof hints, including *loop invariants*, *assertions*, and *lemma functions*, which guide the verifier. Figure 1a presents a running Verus program. Verus operates as an Automated Theorem Proving (ATP) system: it translates the program into SMT formulas and, with the assistance of user-provided proof hints, checks whether the Rust code satisfies the specifications using the SMT solver Z3 (De Moura & Bjørner, 2008).

To the best of our knowledge, existing work on LLMs for Verus focuses *exclusively* on automatic proof-hint synthesis. Systems such as AlphaVerus (Aggarwal et al., 2024), SAFE (Chen et al., 2024), AutoVerus (Yang et al., 2024), RagVerus (Zhong & Si, 2025), and VeriStruct (Sun et al., 2025) specialize LLMs for generating proof hints, while VeruSAGE (Yang et al., 2025a) evaluates model performance on this task. However, existing approaches treat the verification process as a ***black box***, measuring success solely by a ***binary*** outcome: whether the program verifies with the generated proof hints.

However, such binary evaluation obscures the central question: ***Can LLMs systematically reconstruct the verification reasoning, or are they merely exploiting statistical patterns?*** Judging a model solely by whether a program verifies reveals nothing about ***how*** verification was achieved. It cannot distinguish a model that constructs a sound deductive chain from one that succeeds through chance syntactic alignment. To meaningfully assess whether LLMs exhibit reasoning capabilities comparable to automated theorem provers (e.g., Z3 SMT solver), we must open the black box and examine whether they can construct a **Verifica-**

```
1  fn replace_last_element(first: &Vec<i32>, second: &Vec<i32>)
     -> (replaced_list: Vec<i32>)
2      requires first.len() > 0,
3      ensures replaced_list@ == first@.subrange(0, first.len() -
   1).add(second@),
4  {
       ...
11     while index < first_end
12         invariant
13             first_end == first.len() - 1,
14             0 <= index <= first_end,
15             replaced_list@ =~= first@.subrange(0, index as int),
16     {
17         replaced_list.push(first[index]);
18         index += 1;
19     }
20     index = 0;
21     while index < second.len()
22         invariant
23             0 <= index <= second.len(),
24             replaced_list@ =~= first@.subrange(0, first.len() - 1)
25             .add(second@.subrange(0, index as int),
26             ),
27     {
28         replaced_list.push(second[index]);
29         index += 1;
30     }
31     assert(replaced_list@ =~= first@.subrange(0,
32         first.len() - 1).add(second@));
33     replaced_list
34 }
```

```
...
491  (let ((@x11318 (unit-resolution ((_ quant-inst
     fuel%vstd!seq.axiom_seq_subrange_len.) (or (not $x30) $x11316))
     @x11044 (hypothesis (not $x11316)) false)))
...            High Level: Prove via a proof lemma
622  (let ((@x1995 (monotonicity (rewrite (= $x1925 (and $x1083
     $x1936))) (rewrite (= $x1927 $x1943)) (= $x1928 (and (and
     $x1083 $x1936) $x1943)))))
...         Medium Level: Prove array index is within bounds
653  (let ((?x3594 (lambda ((x Int) (y Int) )(let (($x775 (= (+ y x
     (* (- 1) (Add x y))) 0)))
654  (refl (~ $x775 $x775))))
...           Low Level: Prove $x775 = $x775
879  (let ((@x12791 (symm @x12785 (= ?x12350 ?x10590))))
880  (let (($x12788 (= ?x12401 ?x12350)))
881  (let (($x12786 (= ?x10601 ?x12349)))
...  Low Level: Prove ?x12350 = ?x10590 given ?x10590 = ?x12350
2614 (let ((?x3391 (lambda ((x Int) )(refl (~ (= x (%I (I x))) (= x
     (%I (I x))))))
...    Low Level: Prove that for any integer x, x = x as int
5335 (let (($x13538 (= ?x11276 ?x12460)))
5336 (let (($x14221 (not $x13538)))
5337 (let ((@x14218 (commutativity (= $x13538 $x12914))))
...Low Level: Prove ($x13538 = $x12914) <==> ($x12914 = $x13538)
7793 (let (($x2919 (= $x2915 $x2918)))
7794 (refl (~ $x2919 $x2919)))))))))))
...             Low Level: Prove $x2919 = $x2919
10,003 lines in total
```

*(a)* A running example of a Verus program.      *(b)* The Z3 proof for the example program.

*Figure 1.* (a) The program is about replacing the last element of the first vector (`first`) with all elements of the second (`second`). Executable Rust code, specifications, and proof hints are highlighted. The specifications define the precondition (`requires`) and postcondition (`ensures`), and verification relies on three parts of proof hints: two blocks of loop invariants and one assertion. (b) Shows the Z3 proof of the running example with high-level, medium-level, and low-level proofs highlighted. The high-level rule `unit-resolution` (line 491) establishes the property $|seq.subrange(s, j, k)| = k - j$, while medium-level rules such as `monotonicity` and `rewrite` (line 622) prove index bounds. Low-level rules encode trivial reasoning, e.g., `refl` (lines 653–654), which simply asserts `$x775 = $x775`.

**tion Chain-of-Thought (VCoT)**, the step-by-step deductive process underlying formal verification, analogous to chain-of-thought reasoning in natural language.

Opening the verification black box is fundamentally hindered by the nature of the ATP solver. When verification succeeds, Z3 can emit a proof trace, but these traces are designed for machine consumption rather than human understanding. As shown in Figure 1b, the Z3 proof for our small running example spans 10,003 lines, yet 81.69% of it consists of low-level reasoning, such as trivial equalities like `$x775 = $x775`. This extreme verbosity creates a semantic gap that obscures the solver's internal reasoning, leaving existing research confined to a results-oriented paradigm that cannot inspect, analyze, or evaluate the underlying verification process.

To bridge this gap, we introduce **VCoT-Lift**, an LLM-based framework that lifts low-level Z3 solver reasoning into explicit, human-readable Verus verification steps, exposing the solver's internal reasoning as a VCoT and providing transparent ground truth for program verification. Constructing such a VCoT is fundamentally a semantic abstraction problem, raising three core challenges: **soundness**, **completeness**, and **conciseness**. VCoT-Lift addresses these challenges through a four-stage pipeline that enforces soundness via direct interaction with the Verus verifier, improves

completeness through an interactive loop between global transformation and targeted completeness checking, and enhances conciseness by principled pruning of trivial and redundant reasoning.

Leveraging VCoT-Lift, we introduce **VCoT-Bench**, a comprehensive benchmark that moves LLM evaluation beyond binary pass/fail outcomes to a fine-grained analysis of verification reasoning. VCoT-Bench consists of 1,988 VCoT completion tasks constructed by digging structured "proof holes" across the entire VCoT at the level of *semantic blocks*. The benchmark evaluates models' ability to reconstruct verification reasoning along **three orthogonal dimensions**: 1) robustness to varying degrees of missing reasoning; 2) competence across proof types; and 3) sensitivity to the position of missing reasoning steps.

We conducted a comprehensive **study** of ten state-of-the-art LLMs using VCoT-Bench, revealing a substantial gap between LLMs and automated theorem provers. While LLMs can often complete verification steps given sufficient local context, their performance drops sharply even under modest information loss, exposing a heavy reliance on syntactic scaffolding rather than principled deduction. This weakness is most pronounced in intermediate, connective proof steps that require tracking program state and composing multistep reasoning, as well as in assertion proofs demanding

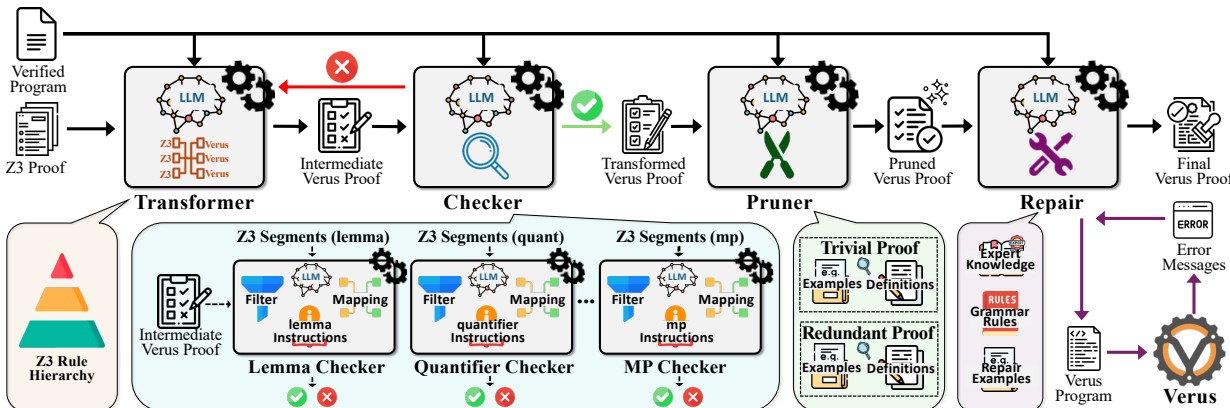

*Figure 2.* Overview of VCoT-Lift.

precise, state-dependent logic.

In summary, this paper makes the following contributions:
- **Concept**: We propose the new concept of VCoT.
- **Tool**: We develop VCoT-Lift, the first framework that lifts low-level reasoning into high-level VCoT.
- **Benchmark**: We introduce VCoT-Bench, the first benchmark for VCoT completion, containing 1,988 tasks stratified by three orthogonal evaluation dimensions.
- **Study**: We perform a comprehensive study of ten LLMs, uncovering their critical limitations in formal verification.

Artifacts of VCoT-Lift and VCoT-Bench are available at https://github.com/HIPREL-Group/VCoT-Bench.

## 2. VCoT-Lift

In this section, we introduce VCoT-Lift, an LLM-based framework that lifts low-level reasoning from Z3 proofs into explicit Verus-level verification steps, forming a VCoT that exposes how verification unfolds at the program level.

**Why use LLMs?** Transforming Z3 proofs into Verus-level verification steps is a *semantic abstraction challenge*: Z3 proofs are extremely fine-grained, while Verus operates over high-level constructs such as invariants, assertions, and lemmas, making the required semantic compression beyond static pattern matching or deterministic symbolic translation (Böhme & Nipkow, 2010; Blanchette et al., 2013; Mohamed et al., 2025). LLMs can recover this abstraction by jointly analyzing Z3 proofs and Verus programs to infer logical intent and synthesize equivalent high-level verification steps (Li et al., 2023; Liu et al., 2024; Wei et al., 2025; Wen et al., 2025; Li et al., 2022; Jiang et al., 2024; Islam et al., 2024; Zhuo et al., 2024), and thus VCoT-Lift adopts LLMs as its central reasoning engine.

**Transformation Challenges.** Even with LLMs, transforming Z3 proofs into Verus-level verification steps remains difficult. This process must simultaneously address three fundamental challenges:

- **Soundness**: The transformed verification steps should faithfully preserve the semantics of the original Z3 proof.
- **Completeness**: The transformed verification should capture the entire reasoning process of the Z3 proof, with no essential steps omitted.
- **Conciseness**: The resulting verification steps should be concise, free from trivial or redundant steps.

### 2.1. Overview

To address these challenges, we present VCoT-Lift as a four-stage pipeline (Figure 2). The pipeline starts with an iterative transformer–checker loop that improves proof completeness: the transformer synthesizes Verus proofs from Z3 proofs, and the checker evaluates their completeness until predefined criteria are met. Next, the Proof Pruner removes trivial or redundant steps to improve conciseness. Finally, Proof Repair enforces soundness by interacting with Verus to detect and fix syntactic and semantic errors. We detail each stage and its interactions in the following. Appendix B.2 reports how we choose models to instantiate each stage of VCoT-Lift.

Figure 8 in Appendix A shows the complete VCoT exposed by VCoT-Lift for the running example, including 82 lines of additional lifted Verus proofs, which naturally unfold into five verification phases.

### 2.2. Proof Transformer

Given a Verus program and its corresponding Z3 proof, Proof Transformer aims to transform the internal reasoning of the Z3 proof into semantically equivalent Verus proofs.

The key challenge is that Z3 proofs are long and dominated by low-level reasoning, with critical logical steps tightly entangled throughout. Disentangling this structure requires a holistic view of the entire proof: because the reasoning is globally interconnected, the transformer cannot safely decompose proofs without losing context and must process them end to end to recover the underlying semantics. However, such long-context comprehension remains a major bottleneck for current LLMs (Li et al., 2024; Dong et al.,

```
 ...                                                    ①
37 while index < second.len()
38     invariant
39         0 <= index <= second.len(),
40         replaced_list@ =~= first@.subrange(0, first.len() - 1)
41           .add(second@.subrange(0, index as int),
42         ),
43 {
44     replaced_list.push(second[index]);
45     index += 1;
46 }                                          Iteration 1
47 +proof {
48 +    assert(0 <= index <= second@.len());   Iteration 2
49 +    assert(second@.subrange(0, index) == second@) by {
50 +        lemma_subrange_all(second@);
51 +    }
52 +    assert(replaced_list@ =~= first@.subrange(0,
53 +      first.len() as int - 1).add(second@));
54 +}                                          Iteration 3
55 +proof {
56 +    let target = first@.subrange(0, first.len() - 1)
57 +      .add(second@);
58 +    assert(replaced_list@.ext_equal(target)) by {
59 +        assert(replaced_list@ =~= target);
60 +    }
61 +}
 ...                                          Iteration 1
65 +proof fn lemma_subrange_all<A>(s: Seq<A>)
66 +    ensures
67 +        s.subrange(0, s.len() as int) == s,
68 +{
69 +    assert(s.subrange(0, s.len() as int) =~= s);
70 +}                                    [Transformer]
 ...
47 proof {                                              ②
48 -    assert(0 <= index <= second@.len());
49     assert(second@.subrange(0, index) == second@) by {
50         lemma_subrange_all(second@);
51     }
 ...
55 -proof {
56 -    let target = first@.subrange(0, first.len() - 1)
57 -      .add(second@);
58 -    assert(replaced_list@.ext_equal(target)) by {
59 -        assert(replaced_list@ =~= target);
60 -    }
61 -}                                         [Pruner]
 ...
48 proof {                                              ③
-    assert(second@.subrange(0, index) == second@) by {
49 +    assert(second@.subrange(0, index as int) == second@) by {
50         lemma_subrange_all(second@);
51     }
 ...                                          [Repair]
```

*Figure 3.* The intermediate VCoT for the running example's second while loop across stages of VCoT-Lift. ① Shows the Verus program with original proof hints and transformed proofs; after three transformer–checker loops, several assertions and a lemma establishing a key property s.subrange(0, s.len()) == s are introduced. ② Shows the pruned proofs. ③ Shows the repaired proofs, fixing a syntactic error.

2024; Hosseini et al., 2025; Du et al., 2025).

To address this challenge, we introduce a *Z3 rule hierarchy* that explicitly steers LLM attention toward reasoning steps more likely to be semantically informative. Our classification is based on syntactic cues of proof rules, but is motivated by empirical observations of how often such syntactic patterns tend to surface semantically meaningful reasoning in practice. Concretely, we organize all 36 Z3 proof rules (Z3Prover, 2026) into three importance levels:

- **High-level rules (8)**: frequently correspond to explicit Verus verification steps (e.g., unit-resolution).

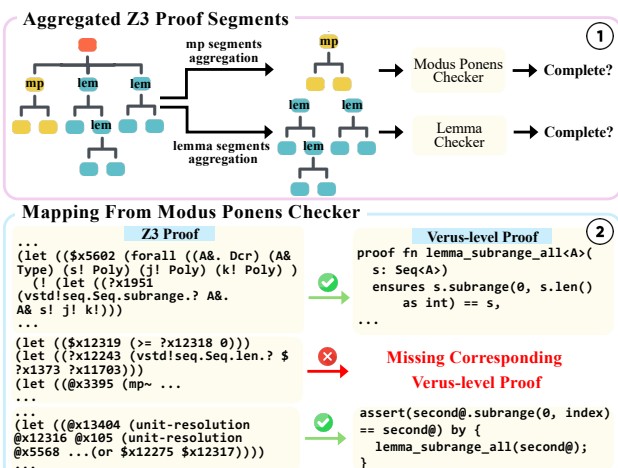

*Figure 4.* Conceptual visualization of: ① Aggregated Z3 proof segments for the running example, where mp and lemma segments are grouped and passed to the corresponding checkers; ② the mapping mechanism, where the modus ponens checker identifies a Z3 proof without a corresponding Verus-level proof and marks the transformation incomplete.

- **Medium-level rules (12)**: support intermediate proof construction (e.g., trans and rewrite).
- **Low-level rules (16):** primarily encode tautologies, syntactic artifacts, or proof metadata (e.g., refl and symm).

Table 2 in Appendix B.1 presents the full categorization. Figure 1b illustrates a Z3 proof of our running example annotated by rule importance. Figure 3 part ① shows the verification steps transformed from the Z3 proof by Proof Transformer.

### 2.3. Proof Checker

The Proof Checker is a key component of VCoT-Lift that evaluates the completeness of proofs produced by the Proof Transformer. To improve efficiency and focus on essential reasoning, completeness checking is restricted to *high-level* Z3 proof rules, verifying only whether critical proof steps are correctly transformed. Because these rules share common reasoning patterns, we group the eight high-level rules into five categories: lemma (lemma), theory lemma (th-lemma), modus ponens (mp, mp~), quantifier (quant-intro, quant-inst, skolemize), and unit resolution (unit-resolution).

To enable precise and specialized completeness assessment, we design a dedicated *checker agent* for each proof-rule category to evaluate the completeness of its corresponding transformations.

**Inputs: Aggregated Z3 Proof Segments.** To enable focused completeness check, each checker agent receives only the Z3 proof segments relevant to the proof rules it assesses. VCoT-Lift constructs an abstract syntax tree of the Z3 proof, where each node represents a proof statement, and its subtree contains all steps required to derive that statement, referred to as the rule's dependency. By locating the node for a target

proof rule and extracting its subtree, VCoT-Lift precisely isolates the proof segments associated with that rule.

Even high-level proof rules occur frequently in Z3 proofs, and processing each segment independently would require many LLM invocations. Moreover, proof segments, especially within the same category, are often interdependent; for example, one lemma may depend on another, as shown in Figure 4, part ①. Evaluating segments in isolation would thus introduce redundant context and unnecessary LLM calls.

To address this issue, VCoT-Lift *aggregates all segments* from the same proof-rule category and provides them as a single input to the corresponding checker agent. As shown in Figure 4, part ①, the lemma checker receives all lemma rules together with their dependencies in one batch, enabling joint assessment and avoiding redundant processing.

**Checking Mechanism 1: Abstraction Filter.** Although essential reasoning is often expressed with high-level proof rules, such rules can also encode trivial or redundant steps. For example, the Z3 proof `unit-resolution @x11036 @x3395 $x11026` uses the high-level rule `unit-resolution` but merely establishes `0 = 0`. A checker agent may mistakenly treat such trivial steps as essential missing logic, causing it to diverge (e.g., fail to terminate) by repeatedly flagging the proof as incomplete based on unnecessary reasoning.

To address this issue, VCoT-Lift introduces an *abstraction filter* that regulates proof granularity and guides the checker to ignore unnecessary reasoning. It distinguishes between *trivial* and *redundant* proof steps, both of which can be safely excluded from completeness evaluation.

*Trivial steps* correspond to *local reasoning* that the solver can justify using only immediate contextual facts, without invoking additional axioms or unfolding complex definitions. Typical examples include basic arithmetic facts and properties such as reflexivity (e.g., `index = index`) and commutativity (e.g., $((x + 0) = x) \leftrightarrow (x = (x + 0))$).

*Redundant steps* are reasoning steps that add no new semantic information beyond what is already established. We categorize redundancy into three forms: (1) normalization, which rewrites expressions into canonical forms; (2) reassertion, which restates facts already implied by the current context; and (3) definition expansion, which unfolds definitions without advancing the core proof argument.

In our running example, the modus ponens checker initially flags `index < index + 1` as a missing proof step, but it is correctly classified as trivial and filtered out by the abstraction filter, allowing the proof to be marked complete.

**Checking Mechanism 2: Explicit Mapping.** To reduce hallucinations and strengthen reasoning, VCoT-Lift adopts

an explicit justification mechanism. Instead of issuing a binary completeness verdict, each checker must justify its decision by *explicitly mapping* each Z3 proof step within its rule category to a semantically equivalent step in the transformed Verus proof. Figure 4, part ②, illustrates this mapping mechanism, where the modus ponens checker explicitly identifies that the step `assert(0 <= index <= second@.len())` has no corresponding transformation. This structured, traceable output improves checker reliability and enables rigorous human validation.

**Checking Mechanism 3: Specialization.** For each checker agent, we describe the semantics and usage of the associated Z3 proof rules and specify the forms of Verus proofs into which they are commonly transformed. For example, for the lemma checker, we explain that lemmas discharge hypotheses and that a `lemma` proof rule is often transformed into a lemma function, a proof block, or a sequence of assertions.

## 2.4. Transformer–Checker Loop

VCoT-Lift further addresses extremely long proof contexts with a *perform-all-check-partial* architecture that deliberately separates responsibilities: the Proof Transformer operates with a global view of the entire Z3 proof, while the checker validates only a targeted subset of high-level, essential proof steps.

To preserve this asymmetry, the checker emits only coarse-grained binary signals (complete or incomplete), without explanations. Because its perspective is inherently partial, providing localized feedback could mislead or constrain the transformer's global reasoning. By acting as lightweight guardrails, the checkers guide transformation without polluting the transformer's context.

This separation improves scalability and strengthens transformation robustness. When the checker signals incompleteness, the transformer reprocesses the entire Z3 proof holistically, naturally recovering additional missing steps, including those beyond the checker's limited scope. In this way, VCoT-Lift combines efficient partial checking with global regeneration to achieve comprehensive and reliable proof transformation under long-context constraints.

In Figure 3, part ①, across three transformer–checker loops, the proofs become progressively more complete.

## 2.5. Proof Pruner

The transformed Verus proofs may include trivial or redundant steps. For example, handling `a % 2` can introduce a trivial assertion `assert(2 != 0)` to prevent division by zero. While such checks are reasonable at the Z3 level, they are unnecessary in Verus. To improve conciseness, VCoT-Lift introduces a Proof Pruner to remove these trivial and redundant steps.

Building on the abstraction filter's definitions of trivial and

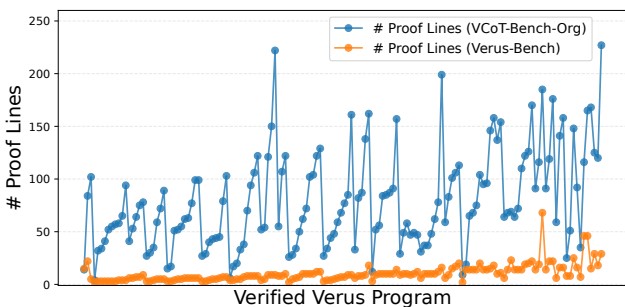

*Figure 5.* Comparison of Total Proof Lines per Program Between VCoT-Bench-Org and Verus-Bench

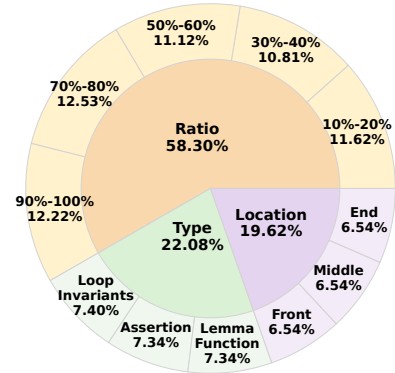

*Figure 6.* Composition of VCoT-Bench.

redundant steps, the Proof Pruner refines these notions with fine-grained criteria tailored to the three Verus proof types, supported by representative examples. For instance, redundant assertions restate facts already established in nearby proof steps. To preserve rigor, the pruner agent must provide an explicit justification for every removal, ensuring the resulting proofs are both concise and transparent for human validation. As shown in Figure 3, the assertion `assert(0 <= index <= second@.len())` is pruned because it is already established by the preceding `while` loop invariants (line 39); and a proof block (lines 55–61) is removed since it duplicates the preceding assertion (lines 52–53).

### 2.6. Proof Repair

Proof Repair is the final stage of the pipeline and ensures the soundness of the transformed verification steps. Because Verus is relatively new and has limited public resources, LLMs often produce syntactic or semantic errors during transformation. To address this, VCoT-Lift introduces a dedicated repair agent that systematically detects and corrects such errors.

To mitigate model unfamiliarity with Verus grammar, we inject expert verification knowledge to guide the repair agent. For example, when multiple type conversions cause misleading error messages, the agent is instructed to explicitly parenthesize each conversion. We also provide explicit grammar rules and curated repair examples. At this stage, the Verus verifier compiles and checks the program with the transformed proofs, and the resulting error messages are fed back to the repair agent, which iteratively refines the proofs until successful verification. As shown in Figure 3, in our running example, Proof Repair corrects a syntactic error for a transformed proof step.

## 3. VCoT-Bench

Leveraging VCoT-Lift to construct VCoTs as ground truth, we create VCoT-Bench, a benchmark that rigorously evaluates whether LLMs can systematically reconstruct the verification process.

### 3.1. Ground-Truth: VCoT-Bench-Org

We build our benchmark on Verus-Bench (Yang et al., 2024), a widely used public benchmark for Verus proof-hint generation. Verus-Bench comprises 150 verified Verus programs drawn from MBPP-DFY-153 (Misu et al., 2024), Clover-Bench (Stack Overflow, 2024), Diffy (Chakraborty et al., 2021), and verified algorithms from the Verus libraries. Figure 1a shows an example from Verus-Bench (task_id_240).

For each verified program, we extract its Z3 proof and apply VCoT-Lift to lift low-level reasoning into explicit Verus-level verification steps, yielding the program's VCoT. This process produces VCoT-Bench-Org, a ground-truth dataset with fully expanded verification reasoning. As shown in Figure 5 and Figure 9 in Appendix B.3, VCoT-Bench-Org exposes substantially richer verification details than Verus-Bench: on average, it exhibits a 6.5x increase in proof lines (up to 32x), a 13.4x increase in assertions (up to 54x), and a 1.94x increase in lemma functions (up to 9x). The spikes in Figure 5 reflect variability in verification complexity across programs: simple programs such as single-loop examples require relatively few lifted steps, whereas complex programs with nested loops or recursive lemmas require many more intermediate proof steps after VCoT-Lift lifting. The cost for constructing VCoT-Bench-Org is reported in Appendix B.3.

### 3.2. VCoT Completion Tasks via Semantic Blocks.

To assess how well LLMs can reconstruct VCoT steps, we construct VCoT completion tasks by digging "proof holes" in VCoT-Bench-Org at the granularity of *semantic blocks*. Each semantic block falls into one of three categories: *Lemma Blocks*, where each independent lemma function forms a block; *Invariant Blocks*, where all loop invariants associated with the same loop form a block; and *Assertion Blocks*, where all assertions between two executable code lines are grouped into one block. As shown in Figure 3, Part ①, the lemma function `lemma_subrange_all` (lines 65–70) forms one semantic block; the loop invariants associated with the `while` loop (lines 38–42) form another; and the assertions after the `while` loop constitute the third block (lines 47–61).

### 3.3. Three Benchmark Variants

With the semantic blocks, we construct the full VCoT-Bench suite with 1,988 VCoT completion tasks. To rigorously evaluate LLMs' ability to reconstruct the entire verification process, we organize these tasks along three orthogonal dimensions, *Ratio*, *Type*, and *Location*. Detailed statistics of VCoT-Bench are provided in Figure 6.

**VCoT-Bench-Ratio.** This sub-benchmark evaluates an LLM's ability to recover missing verification steps under varying degrees of information loss, quantified by the ratio of removed semantic blocks. For each program in VCoT-Bench-Org containing $N$ semantic blocks, we generate $N$ variants by randomly removing between 1 and $N$ blocks and computing the corresponding removal ratios. This procedure yields a total of 1,159 VCoT completion tasks.

**VCoT-Bench-Type.** This sub-benchmark evaluates an LLM's ability to reconstruct specific proof types. For each program, we generate up to three variants by removing all blocks of a given type: *Invariant-removal* (all loop invariants), *Assertion-removal* (all inline assertions), and *Lemma-removal* (all lemma functions). This procedure yields 439 VCoT completion tasks.

**VCoT-Bench-Loc.** The order of the VCoT is critical. This sub-benchmark evaluates how the position of missing reasoning steps affects a model's ability to recover the underlying logic. We define three removal zones: *Front* (first 33% of the VCoT), *Middle* (33%–66%), and *End* (final 33%). This process yields 390 VCoT completion tasks.

## 4. Study

Using VCoT-Bench, we conduct a thorough study of ten state-of-the-art LLMs, evaluating their verification capabilities by measuring performance on VCoT completion.

### 4.1. Study Setup

**Subjects.** We study ten state-of-the-art LLMs in a zero-shot setting, comprising six proprietary models: GPT-5.2, GPT-5 mini, Claude Sonnet 4.5, Claude Haiku 4.5, Gemini 3.1 Pro, and Gemini 3 Flash; and four open-source models: DeepSeek V3.2 (685B), DeepSeek R1 (685B), Qwen 3 (8B, evaluated under both thinking and non-thinking inference modes), and gpt-oss (20B). All models are run with default settings, with output limits set sufficiently high for all programs. Table 3 in Appendix B.4 summarizes their configuration details.

**Decoding Strategy.** We evaluate both greedy decoding and Best-of-10 sampling. Due to space constraints, we report only greedy-decoding results, while detailed Best-of-10 sampling results are provided in Appendix D.2. Additional evaluation details, including temperature settings, context/output limits, and semantic-judge costs are reported in Appendix B.5.

**Evaluation Metrics.** To evaluate LLM performance on VCoT completion tasks, we employ three metrics: ***Syntactic Accuracy (SynAcc)***, ***Semantic Accuracy (SemAcc)***, and ***Overall Accuracy (Acc)***.

Syntactic correctness is checked by invoking Verus in syntax-only mode (`-no-verify`). Semantic correctness is evaluated by a specialized LLM-based judge, guided by a detailed protocol with curated examples to align its decisions with human judgment. We use GPT-5 mini as the final judge. Specifically, we randomly sample 100 stratified tasks from VCoT-Bench and evaluate eight judge candidates, covering the strongest LLMs available at submission time from four model families: OpenAI, Claude, Gemini, and DeepSeek. Each judge is evaluated on outputs from all ten models, with accuracy measured against author consensus. Detailed results are reported in Appendix D.1. GPT-5 mini achieves the highest overall accuracy (92.7%) and is therefore selected as the semantic judge.

For overall accuracy, inspired by CodeBLEU (Ren et al., 2020), we define a weighted accuracy metric that emphasizes semantic correctness over syntactic correctness. Each prediction is categorized into one of four levels:

- *Level 3:* Both semantically and syntactically correct.
- *Level 2:* Semantically correct but syntactically incorrect.
- *Level 1:* Syntactically correct but semantically incorrect.
- *Level 0:* Incorrect in both dimensions.

Overall accuracy is computed as

$$\text{Acc} = \frac{N^{(3)} + 0.5\,N^{(2)} + 0.25\,N^{(1)}}{N} \times 100\%, \quad (1)$$

where $N^{(i)}$ denotes the number of cases at Level $i$, and $N$ is the total number of cases.

**Notes:** Due to space constraints, we report and analyze only overall accuracy. Detailed results and discussion of syntactic and semantic accuracy are deferred to Appendix C.

### 4.2. Performance across Proof-Removal Ratios.

Using VCoT-Bench-Ratio, we evaluate LLM performance across varying proof-removal ratios and identify five key findings from the results shown in Figure 7.

*F1: Context removal exposes fragile reasoning.* Accuracy drops sharply as proof blocks are removed. With only 10% of blocks missing, the best model, Claude Sonnet 4.5, achieves just 71.58% accuracy, while the weakest, gpt-oss, reaches only 32.89%. When all blocks are removed (100%), turning the task into full VCoT construction, performance collapses: Claude Sonnet 4.5 falls to 17.22% and Qwen 3 think to near zero (0.66%). This shows that current LLMs do not reason from first principles but rely heavily on local pattern matching and syntactic scaffolding; once this context

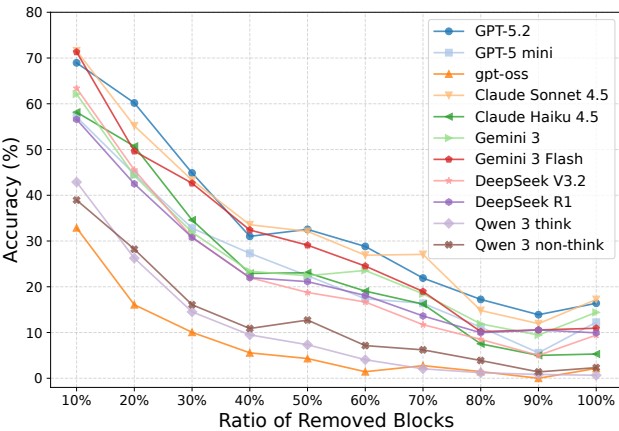

*Figure 7.* Overall accuracy across proof-removal ratios.

*Table 1.* Overall accuracy across proof types and proof locations. The **highest** and second-highest accuracies are highlighted.

| | Results for overall accuracy | | | | | |
|---|---|---|---|---|---|---|
| **Model Name** | **VCoT-Bench-Type** | | | **VCoT-Bench-Loc** | | |
| | Invariant | Assertion | Lemma | Front | Middle | End |
| GPT-5.2 | 54.42 | 34.25 | **55.48** | 58.27 | 41.15 | 65.58 |
| GPT-5 mini | 23.47 | 32.88 | 36.30 | 49.04 | 36.92 | 60.00 |
| gpt-oss | 17.69 | 21.58 | 26.20 | 19.81 | 9.81 | 28.65 |
| Claude Sonnet 4.5 | **68.71** | **39.55** | 51.54 | **69.04** | 46.35 | 65.96 |
| Claude Haiku 4.5 | 52.55 | 27.23 | 42.64 | 58.85 | 35.19 | 57.69 |
| Gemini 3.1 Pro | 62.07 | 22.95 | 42.81 | 62.69 | 36.15 | 46.73 |
| Gemini 3 Flash | 57.65 | 28.60 | 41.61 | 67.12 | **47.69** | 62.50 |
| DeepSeek V3.2 | 49.49 | 29.45 | 44.35 | 56.73 | 32.31 | 58.85 |
| DeepSeek R1 | 46.60 | 28.94 | 38.53 | 53.46 | 27.88 | 48.08 |
| Qwen 3 think | 10.03 | 11.47 | 22.43 | 22.88 | 14.81 | 35.58 |
| Qwen 3 non-think | 26.87 | 13.70 | 23.63 | 33.46 | 16.54 | 37.12 |

is stripped away, they fail to construct complex, multi-step deductive chains.

***F2: A 40% removal threshold breaks proof inference.*** Although accuracy declines overall, the drop is non-uniform. Across all models, performance falls sharply as block removal increases from 10% to 40%, then degrades much more slowly from 40% to 100%. This pattern reflects the loss of critical structural anchors in the verification logic: once these anchors fall below a threshold, logical continuity collapses, and models can no longer infer the proof logic, making it uniformly difficult regardless of further removal.

***F3: Model scale helps.*** Performance is strongly influenced by model scale: larger models (e.g., proprietary variants and DeepSeek V3.2/R1) substantially outperform smaller ones (e.g., gpt-oss and Qwen 3), underscoring that sufficient parameter capacity is crucial for capturing the intricate logical dependencies of verification.

***F4: Reasoning variants can hurt performance.*** We observe counterintuitive inversions within model families: Qwen 3 generally outperforms its "thinking" variant, and Gemini 3 Flash overall surpasses Gemini 3.1 Pro. This suggests that current reasoning-oriented paradigms can inject noise into verification. In formal proofs, where syntactic precision and solver fidelity are critical, verbose "thinking" can induce hallucinated predicates or semantic drift, while simpler models may benefit from a more direct, lower-entropy pattern-matching approach.

To further examine this phenomenon, we inspect CoT traces from failed Qwen 3 think cases and identify two recurring failure patterns. First, CoT often begins with a reasonable high-level proof strategy but gradually deviates during extended reasoning. For example, in loop-invariant reconstruction, the model may correctly identify the relevant sequence properties, but then introduce unnecessary side conditions or subtly alter predicates, breaking semantic equivalence with the target proof. This suggests that longer reasoning chains can increase semantic drift: each additional reason-

ing step creates another opportunity to move away from the verifier-aligned proof obligation.

Second, the model sometimes generates plausible but invalid Verus elements, such as nonexistent lemmas. Once introduced, these hallucinated elements propagate through the remaining chain and lead to syntactic or semantic errors. In contrast, the non-thinking variant tends to stay closer to local contextual patterns and is therefore less exposed to these cascading deviations. These behaviors indicate a mismatch between generic CoT-style reasoning and formal verification. While CoT can help tasks that benefit from open-ended decomposition, VCoT completion requires strict alignment with the verifier's logical structure; additional reasoning steps may therefore introduce noise rather than useful decomposition.

***F5: 100% removal accuracy is inflated by small programs.*** We observe a slight accuracy increase for most models at 100% removal. This is due to the discrete nature of the removal rate. Single-block programs appear only at 100%, causing small, easier programs to be overrepresented and inflating accuracy at this extreme.

### 4.3. Performance across Proof Types

We evaluate LLM performance across proof types using VCoT-Bench-Type and report three key findings based on the results in Table 1.

***F1: Assertions are the hardest.*** Across most models, assertion completion has the lowest accuracy. Even the strongest model, Claude Sonnet 4.5, achieves only 39.55% on assertions, compared to 68.71% on loop invariants and 51.54% on lemma functions. This consistent gap reflects a fundamental weakness: assertions require precise, state-specific reasoning where missing a single condition causes failure, whereas invariants and lemmas allow greater reliance on higher-level structural patterns.

***F2: Loop invariants are easiest yet most discriminative.*** For nearly all models, loop invariant completion achieves the highest accuracy, but also shows the widest spread

across models, ranging from 68.71% (Claude Sonnet 4.5) to 10.03% (Qwen 3 think), a 58.68% gap. This variance far exceeds that of assertions (28.08%) and lemma functions (33.05%), indicating that loop invariants are the most discriminative proof type: they require holistic reasoning over iterative program behavior, clearly separating models with genuine verification capability from those relying on shallow pattern matching.

***F3: Model scale helps, but is not decisive.*** Model size generally correlates with higher accuracy, but the trend is not absolute. For example, GPT-5 mini (23.47%) underperforms Qwen 3 non-think (26.87%) on loop invariants. While proprietary models typically lead, DeepSeek V3.2 matches the performance of Claude Haiku 4.5, showing that well-trained open-source models can narrow the gap.

### 4.4. Performance across Proof Locations

We evaluate LLM performance across proof locations using VCoT-Bench-Loc and report two key findings based on the results in Table 1.

***F1: Connective reasoning in Middle is the hardest.*** Across all strong models, performance drops most sharply when missing blocks occur in the Middle of the proof. For example, Claude Sonnet 4.5 falls from 69.04% (Front) to 46.35% (Middle), and Gemini 3.1 Pro from 62.69% to 36.15%. This consistent decline exposes a shared weakness: while models handle Front blocks that set up constraints and End blocks that follow recognizable closing patterns, they struggle with the Middle, where connective reasoning is required. These intermediate steps demand propagating invariants, tracking state evolution, and composing multi-step deductions, tasks for which local pattern matching offers little support.

***F2: Model scale may not benefit Middle.*** Larger models perform well on Front and End blocks but do not dominate on Middle blocks. Gemini 3 Flash achieves the highest Middle accuracy (47.69%), outperforming both Gemini 3.1 Pro (36.15%) and GPT-5.2 (41.15%). This inversion indicates that Middle-block performance is not driven by model scale; instead, models optimized for low-latency reasoning may better track intermediate verification states, while larger, more verbose models are more prone to semantic drift.

## 5. Discussion

Our results show that current LLMs remain far from reliably reconstructing such reasoning. Even the strongest models degrade sharply as proof context is removed, indicating that they often rely on local proof scaffolding rather than constructing deductive chains from first principles. This finding is important because existing proof-generation evaluations may obscure this weakness: a model can sometimes make a verifier pass without demonstrating the ability to faithfully reconstruct the verification process. In contrast, VCoT completion directly evaluates whether a model can recover the missing logical steps within a partially observed proof trajectory. Beyond evaluation, VCoT provides a concrete path toward improving proof-generation models. A VCoT is richer than a final verified program because it records what needs to be proved, how proof steps depend on one another, and why particular invariants, assertions, or lemmas are sufficient for verification. This makes VCoT a natural supervision source for future supervised fine-tuning and reinforcement learning. However, effective training should be guided by diagnosis rather than simply scaling generic proof data. VCoT-Bench provides this diagnosis by showing that model failures are structured, not random.

- VCoT-Bench-Ratio reveals how models depend on context. The sharp performance drop under increasing proof removal shows that models rely heavily on local scaffolding rather than robustly reconstructing deductive chains. This finding motivates curriculum-style training in which proof context is progressively removed, forcing models to internalize verification logic rather than merely copy nearby proof patterns.
- VCoT-Bench-Type reveals what models struggle to reason about. Assertions are consistently harder than loop invariants and lemma functions, indicating that precise, state-dependent reasoning is a key bottleneck. This gap indicates that the main bottleneck lies in precise, state-dependent reasoning rather than only high-level structural recognition. Future training should therefore emphasize assertion-heavy proof segments and use proof-type-aware objectives instead of treating all proof hints uniformly.
- VCoT-Bench-Location reveals where models fail within a proof. Middle proof blocks are especially challenging because they require connective reasoning. This suggests that reinforcement learning for proof generation should reward correct intermediate reasoning steps, not only final verifier success, and that training data should emphasize proof segments that connect different phases of a verification trajectory.

## 6. Conclusion

This paper introduces VCoT-Lift and VCoT-Bench, which expose solver-level reasoning as explicit VCoTs to rigorously assess whether LLMs can reconstruct the step-by-step logic of Rust verification, revealing a wide gap between current models and automated theorem provers in genuine deductive reasoning. Looking forward, VCoT can go beyond evaluation, providing a concrete supervision signal for training, diagnosing, and guiding learning-based verification systems toward symbolic alignment, and reframing the field from whether models pass verification to whether they can truly reconstruct and reason through the underlying verification logic.

## Acknowledgments

We thank Lingming Zhang and Chenyuan Yang for valuable discussions and ideas during the preliminary stage of this work. We also thank Omar Muhammad for helpful assistance during the project.

## Impact Statement

This paper presents work whose goal is to advance the field of machine learning. There are many potential societal consequences of our work, none of which we feel must be specifically highlighted here.

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

## A. Complete VCoT Exposed by VCoT-Lift

Figure 8 shows the complete VCoT exposed by VCoT-Lift for the running example program, which replaces the last element of the first vector (`first`) with all elements of the second vector (`second`). The VCoT is composed of five primary reasoning phases.

**Phase 1: Preparing sequence properties** Before the first loop, the proof establishes foundational sequence properties needed later by invoking auxiliary lemmas on sequence concatenation and subranges. `lemma_subrange_add_whole` shows that splitting a sequence into two parts and then concatenating them reconstructs the original sequence, while `lemma_seq_add_append_one` shows that, during sequence construction, appending an element commutes with concatenation. Together, these lemmas ensure correct splitting and recombination of sequences and provide the algebraic basis for the loop invariants.

**Phase 2: Verifying the first loop.** The first loop copies all but the last element of `first` into `replaced_list`. Its loop invariant maintains that `replaced_list` equals `first@.subrange(0, index)`. Each iteration appends `first[index]`, and the invariant is preserved by showing that extending the subrange by one element matches the effect of `push`.

**Phase 3: Bridging between the two loops.** After the first loop terminates, the proof establishes that `replaced_list` equals `first.subrange(0, first.len() – 1)`. This result is then normalized to match the second loop invariant by rewriting it as an addition with an empty subrange of `second`, namely (`first@.subrange(0, first.len() – 1).add(second@.subrange(0, 0))`).

**Phase 4: Verifying the second loop.** The second loop appends all elements of `second` to `replaced_list`. Its loop invariant states that `replaced_list` equals `first@.subrange(0, first.len() – 1).add(second@.subrange(0, index))`. Each iteration extends the subrange of `second` by one element, and the invariant is preserved by reasoning that `push` corresponds to subrange extension.

**Step 5: Finalizing the verification.** After the second loop finishes with `index == second.len()`, a final proof step applies lemma function `lemma_subrange_all` to prove `second@.subrange(0, second.len())` equals `second`. This establishes the postcondition `replaced_list@ == first@.subrange(0, first.len() – 1).add(second@)`, completing the verification.

## B. Additional Details

### B.1. Z3 Proof Rule Hierarchy

Table 2 provides a detailed overview of 36 Z3 proof rules. For each rule, we report its importance level along with a detailed explanation of its semantics and usage.

### B.2. VCoT-Lift Pipeline Details

In VCoT-Lift, we use GPT-5.2 for the Proof Transformer, Proof Checker, and Proof Pruner, and Gemini 3.1 Pro for Proof Repair. We select these models based on an ablation over 20 randomly sampled Verus-Bench programs, comparing GPT-5.2, Gemini 3.1 Pro, and Claude Sonnet 4.5 at each stage. For Z3-to-Verus reasoning stages, including transformation, checking, and pruning, we measure the number of lifted proof lines, since more lifted lines increase the chance of recovering a complete Verus proof. GPT-5.2 performs best, producing 45 proof lines on average, compared with 21 for Gemini 3.1 Pro and 7 for Claude Sonnet 4.5. For Verus syntactic and semantic refinement, we measure average repair rounds. Gemini 3.1 Pro performs best with 4 repair rounds on average, followed by Claude Sonnet 4.5 with 8 rounds and GPT-5.2 with 13 rounds.

### B.3. VCoT-Bench

We provide additional details comparing the numbers of assertions and lemma functions between VCoT-Bench-Org and Verus-Bench, as shown in Figure 9.

As shown in Figure 9a, programs in VCoT-Bench-Org contain substantially more assertions, which are a core component of VCoT because they explicitly expose intermediate verification reasoning. Each program contains an average of 15.47 additional assertions, with a maximum of 56. Moreover, among 150 programs, 145 (96.67%) show an increase in the number of assertions.

Figure 9b compares the numbers of lemma functions, showing that programs in VCoT-Bench-Org also include significantly more lemmas, whereas only a small fraction of Verus-Bench programs contain lemma functions. Each program contains an average of 1.94 additional lemma functions, with a maximum of 9. Furthermore, 123 out of 150 programs (82%) show an increase in the number of lemma functions.

We omit a detailed comparison of loop invariant counts because their increase is marginal (1.99%). Unlike assertions and lemma functions, which depend on one another to form a chain of thought, Z3 reasons about loop invariants as a fixed point. As a result, loop invariants act as a summarized representation of a loop, with limited additional verification information to be lifted from the Z3 proof.

```
1   #[verifier::exec_allows_no_decreases_clause]
2   fn replace_last_element(first: &Vec<i32>, second: &Vec<i32>)
    -> (replaced_list: Vec<i32>)
3     requires
4       first.len() > 0,
5     ensures
6       replaced_list@ == first@.subrange(0, first.len() -
    1).add(second@),
7   {
8     let mut replaced_list = Vec::new();
9     let first_end = first.len() - 1;
10    let mut index = 0;
                                                    Phase 1
12  + proof {
13  +   let empty_s: Seq<i32> = Seq::empty();
14  +   if first.len() > 0 {
15  +     lemma_seq_add_append_one(empty_s, empty_s, first[0]);
16  +   }
17  +   assert(Seq::empty().add(Seq::empty().push(first[0])) =~=
    Seq::empty().push(first[0]));
18  +   lemma_subrange_add_whole(first@, first_end as int);
19  + }
20
21    while index < first_end                       Phase 2
22      invariant
23        first_end == first.len() - 1,
24        0 <= index <= first_end,
25        replaced_list@ =~= first@.subrange(0, index as int),
26    {
27        replaced_list.push(first[index]);
28        index += 1;
29    }
30    index = 0;                                     Phase 3
31  + proof {
32  +   assert(replaced_list@ =~= first@.subrange(0, first_end as
    int));
33  +   assert(replaced_list@ =~= first@.subrange(0, first.len()
    as int - 1));
34  +   assert(replaced_list@ =~= first@.subrange(0, first.len()
    as int - 1).add(second@.subrange(0, 0)));
35  + }
36
37    while index < second.len()                     Phase 4
38      invariant
39        0 <= index <= second.len(),
40        replaced_list@ =~= first@.subrange(0, first.len() - 1)
41          .add(second@.subrange(0, index as int),
42        ),
43    {
44        replaced_list.push(second[index]);
45        index += 1;
46    }
                                                    Phase 5
48  + proof {
49  +   assert(second@.subrange(0, index as int) == second@) by {
50  +     lemma_subrange_all(second@);
51  +   }
52      assert(replaced_list@ =~= first@.subrange(0,
53        first.len() as int - 1).add(second@));
54  + }
55    replaced_list
56  }
57
58  +proof fn lemma_subrange_all<A>(s: Seq<A>)
59  +   ensures
60  +       s.subrange(0, s.len() as int) == s,
61  +{
62  +   assert(s.subrange(0, s.len() as int) =~= s);
63  +}
64
```

```
65  +proof fn lemma_subrange_add_whole<A>(s: Seq<A>, n: int)
66  +   requires
67  +     0 <= n <= s.len(),
68  +   ensures
69  +     s.subrange(0, n).add(s.subrange(n, s.len() as int)) == s,
70  +{
71  +   assert(forall|i: int| 0 <= i < s.len() ==>
72  +     s.subrange(0, n).add(s.subrange(n, s.len() as
    int)).index(i) == s.index(i)) by {
73  +       assert forall|i: int| 0 <= i < s.len() implies
74  +         s.subrange(0, n).add(s.subrange(n, s.len() as
    int)).index(i) == s.index(i)
75  +       by {
76  +         if i < n {
77  +           assert(s.subrange(0, n).add(s.subrange(n, s.len()
    as int)).index(i)
78  +             == s.subrange(0, n).index(i));
79  +           assert(s.subrange(0, n).index(i) == s.index(i));
80  +         } else {
81  +           let j = i - n;
82  +           assert(s.subrange(0, n).add(s.subrange(n, s.len() as
    int)).index(i)
83  +             == s.subrange(n, s.len() as int).index(j));
84  +           assert(s.subrange(n, s.len() as int).index(j) ==
    s.index(n + j));
85  +           assert(n + j == i);
86  +         }
87  +     }
88  +}
89  +   assert(s.subrange(0, n).add(s.subrange(n, s.len() as int))
    == s) by {
90  +       assert(s.subrange(0, n).add(s.subrange(n, s.len() as
    int)) =~= s);
91  +   }
92  +}
93
94  +proof fn lemma_seq_add_append_one<A>(s1: Seq<A>, s2: Seq<A>,
    a: A)
95  +   ensures
96  +     s1.add(s2).push(a) == s1.add(s2.push(a)),
97  +{
98  +   let left = s1.add(s2).push(a);
99  +   let right = s1.add(s2.push(a));
100 +   assert forall|i: int| 0 <= i < left.len() implies
    left.index(i) == right.index(i) by {
101 +     if i < s1.len() {
102 +       assert(left.index(i) == s1.add(s2).index(i));
103 +       assert(s1.add(s2).index(i) == s1.index(i));
104 +       assert(right.index(i) == s1.add(s2.push(a)).index(i));
105 +       assert(s1.add(s2.push(a)).index(i) == s1.index(i));
106 +     } else if i < s1.len() + s2.len() {
107 +       let j = i - s1.len();
108 +       assert(left.index(i) == s1.add(s2).index(i));
109 +       assert(s1.add(s2).index(i) == s2.index(j));
110 +       assert(right.index(i) == s1.add(s2.push(a)).index(i));
111 +       assert(s1.add(s2.push(a)).index(i) ==
    s2.push(a).index(j));
112 +       assert(s2.push(a).index(j) == s2.index(j));
113 +     } else {
114 +       assert(left.index(i) == a);
115 +       assert(right.index(i) == s1.add(s2.push(a)).index(i));
116 +       assert(s1.add(s2.push(a)).index(i) ==
    s2.push(a).index(i - s1.len()));
117 +       assert(i - s1.len() == s2.len());
118 +       assert(s2.push(a).index(s2.len() as int) == a);
119 +     }
120 +   }
121 +   assert(left =~= right);
122 +   assert(left == right);
123 +}
```

*Figure 8.* The complete VCoT exposed by VCoT-Lift for the running Verus example with specifications, including original proof hints and the additional proof steps. The VCoT is composed of five primary reasoning phases.

**Construction cost.** Constructing VCoT-Bench-Org costs $126.66 in total. The transformer–checker loop accounts for 91% of this cost, the pruner accounts for 1.9%, and the repair stage accounts for 7.1%. The three benchmark variants are constructed symbolically with no additional cost.

*Table 2.* Z3 Proof Rule Hierarchy.

| Level | Rule | Description |
|---|---|---|
| High | lemma | Discharge a hypothesis; if a contradiction is derived from an assumption $P$, the rule concludes $\neg P$. |
| | th-lemma | Introduce a theory-specific tautology or conflict clause generated by the solver to justify an inference that is valid within that theory's specific axioms. |
| | mp | Given a proof of $P$ and a proof that $P$ implies $Q$ (or $P = Q$), this rule concludes $Q$. |
| | mp$\sim$ | Given a proof of $P$ and a proof that $P$ is equivalent to $Q$ ($P \iff Q$), this rule concludes $Q$. |
| | unit-resolution | Performs a resolution step between a multi-literal clause and one or more unit clauses. |
| | quant-intro | Given an equivalence between two terms $f(x) \iff g(x)$, concludes that $\forall x.f(x) \iff \forall x.g(x)$. |
| | quant-inst | Justifies the specialized case where a universal quantifier is removed by replacing the bound variable with a specific term, concluding that $(\forall x.\phi) \implies \phi[t/x]$. |
| | skolemize | Eliminates an existential quantifier $\exists x.\phi[x]$ by replacing the bound variable with a fresh constant or function $c$, yielding the equisatisfiable formula $\phi[c]$. |
| Medium | asserted | Introduces an initial formula or axiom provided directly by the user as a premise. |
| | hypothesis | Introduces a temporary assumption into a local scope to facilitate conditional reasoning. |
| | and-elim | Derives a specific conjunct from a conjunction; given a proof of $(l_1 \wedge l_2 \wedge \cdots \wedge l_n)$, concludes a single literal $l_i$, representing the elimination of the logical "and" operator. |
| | not-or-elim | Derives the negation of a specific disjunct from a negated disjunction; given a proof of $\neg(l_1 \vee l_2 \vee \cdots \vee l_n)$, concludes $\neg l_i$, applying De Morgan's laws to extract individual negative literals. |
| | trans | Given proofs of $x = y$ and $y = z$, concludes $x = z$. |
| | trans* | Chains an arbitrary sequence of equality or equivalence steps $(x_1 = x_2, x_2 = x_3, \ldots, x_{n-1} = x_n)$ to conclude the direct relation between the first and last terms $(x_1 = x_n)$. |
| | monotonicity | Given $a_1 = b_1, \ldots, a_n = b_n$, concludes $f(a_1, \ldots, a_n) = f(b_1, \ldots, b_n)$. |
| | der | Simplifies a formula by eliminating a universally quantified variable $x$ when the body contains a conjunct $x = t$ (where $x \notin t$), applying the variable substitution $\phi[t/x]$. |
| | rewrite | Represents a single step of the simplifier where $t_1$ is transformed into $t_2$ such that $\vdash t_1 = t_2$. |
| | rewrite* | Collapses a series of individual simplifications and algebraic reductions into a single proof object, concluding that the initial term is equivalent to the final fully-reduced form. |
| | def-axiom | Introduces a formula that defines a new constant or function symbol. |
| | apply-def | Replaces a defined symbol with its corresponding definition. |
| Low | true | Concludes the logical constant 'true' |
| | =/$\sim$ | Justifies basic term equality or equivalence; it concludes that two terms are identical or logically equivalent ($t \iff t$ or $t = t$). |
| | iff-true | Simplifies a formula by concluding $P \iff \top$ given a proof of $P$ |
| | iff-false | Simplifies a formula by concluding $P \iff \bot$ given a proof of $\neg P$. |
| | goal | Represents the final formula to be proven. |
| | refl | Concludes $t = t$ for any term $t$, serving as the fundamental base case for all equality-based reasoning. |
| | symm | Given a proof of $t_1 = t_2$, concludes $t_2 = t_1$. |
| | commutativity | Given a term $f(x, y)$, concludes $f(x, y) = f(y, x)$ for operations like addition, multiplication, or logical disjunction/conjunction. |
| | pull-quant | Given a formula where a quantifier is nested, such as $(\forall x.\phi) \vee \psi$ (where $x$ is not free in $\psi$), concludes the equivalence $(\forall x.\phi) \vee \psi \iff \forall x.(\phi \vee \psi)$. |
| | push-quant | Given a formula like $\forall x.(\phi \wedge \psi)$, this rule concludes the equivalence $\forall x.(\phi \wedge \psi) \iff (\forall x.\phi) \wedge (\forall x.\psi)$. |
| | elim-unused | Concludes the equivalence $(\forall x.\phi) \iff \phi$ or $(\exists x.\phi) \iff \phi$, where $x$ is not a free variable in $\phi$. |
| | distributivity | Justifies the expansion or factoring of terms where one operator distributes over another; most commonly, it validates identities like $a \times (b+c) = (a \times b) + (a \times c)$ in arithmetic, or $p \wedge (q \vee r) \iff (p \wedge q) \vee (p \wedge r)$ in boolean logic. |
| | nnf-pos | Validates the transformation of a formula into Negation Normal Form (NNF) under a positive context. |
| | nnf-neg | Validates the transformation of a formula into Negation Normal Form (NNF) under a negative context. |
| | iff-oeq | Justifies the equivalence between a formula $P$ and its boolean reduction based on the current context. |
| | def-intro | Justifies the introduction of a new boolean constant $p$ as a proxy for a complex formula $\phi$. |

*Table 3.* Detailed configurations and evaluation costs for the evaluated LLMs

| Vendor | Model Name | Checkpoint | Type | Cost |
|---|---|---|---|---|
| OpenAI | GPT-5.2 (OpenAI, 2025a) | gpt-5.2-2025-12-11 | API | $630.64 |
| | GPT-5 mini (OpenAI, 2025b) | gpt-5-mini-2025-08-07 | API | $86.75 |
| | gpt-oss 20b (OpenAI, 2025c) | gpt-oss-20b | GPU | – |
| Anthropic | Claude Sonnet 4.5 (Anthropic, 2025b) | claude-sonnet-4-5-20250929 | API | $960.99 |
| | Claude Haiku 4.5 (Anthropic, 2025a) | claude-haiku-4-5-20251001 | API | $300.32 |
| Google | Gemini 3.1 Pro (Google, 2026) | gemini-3.1-pro-preview | API | $683.32 |
| | Gemini 3 Flash (Google, 2025) | gemini-3-flash-preview | API | $69.97 |
| DeepSeek | DeepSeek V3.2 (Liu et al., 2025) | DeepSeek-V3.2 | API | $8.73 |
| | DeepSeek R1 (Guo et al., 2025) | DeepSeek-R1-0528 | API | $14.74 |
| Alibaba | Qwen 3 (Team, 2025) | Qwen3-8B | GPU | – |

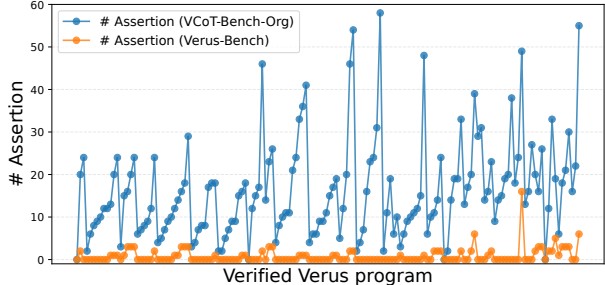

*(a)* Comparison of # Assertions.

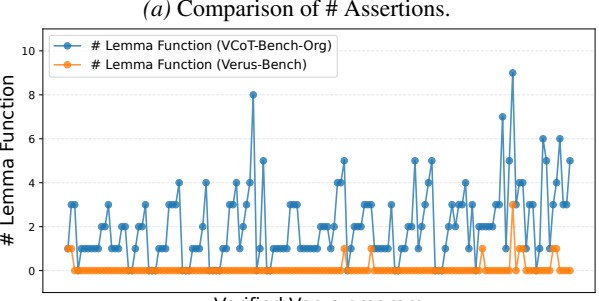

*(b)* Comparison of # Lemma Functions.

*Figure 9.* Comparison of # Assertion and # Lemma Function per program between VCoT-Bench-Org and Verus-Bench.

### B.4. Study Setup

**Machine.** All our experiments are run on a machine with Ubuntu 22.04.5 LTS, 192-core CPUs, and 512GB RAM. For running open-source models locally, we use NVIDIA RTX PRO 6000 Blackwell GPUs with 96 GB VRAM.

**Model Details.** Table 3 presents the details of the evaluated LLMs, including ten models with their respective vendors, model checkpoints, access types (via API or GPU), and evaluation costs.

### B.5. Evaluation Details

**Temperature settings.** We set temperature to 0 for all models to ensure deterministic and reproducible results. The only exception is GPT-5 mini, which does not expose a temperature parameter; for this model, we use its default

API settings.

**Context and output limits.** We impose no additional input limit beyond each model's context window, and all benchmark tasks fit within every evaluated model's context length. We set the maximum output length to 32,768 tokens for all models, and no output reaches this limit.

**Semantic judge cost.** The semantic judge costs $76.27 under greedy decoding and $754.80 under Best-of-10 sampling, for a total judge cost of $831.07.

## C. Analysis of Syntactic and Semantic Accuracy

In this section, we report and analyze the syntactic accuracy (SynAcc) and semantic accuracy (SemAcc) of LLM performance.

### C.1. Performance across Proof Ratios.

Using VCoT-Bench-Ratio, we evaluate LLM performance in both syntax and semantics across semantic proof removal ratios, and report three key findings based on the results shown in Figure 10.

*F1: Proof removal reveals both syntactic and semantic fragility.* Both syntactic and semantic accuracies decline as the proof removal ratio increases. For syntactic accuracy, with 10% removal, the Gemini family achieves near-perfect performance (91.58%), substantially outperforming other models. However, at 100% removal, syntactic accuracy drops sharply to 45.7% for Gemini 3.1 Pro and 24.5% for Gemini 3 Flash. This steep decline indicates that current LLMs lack a robust understanding of Verus syntax and instead rely on local contextual cues, which disappear as more proofs are removed, exposing structural fragility.

For semantic accuracy, with 10% removal, GPT-5.2 achieves the highest performance (73.68%), but this drops to 27.15% at 100% removal. The decline is even more severe for Qwen 3, whose semantic accuracy falls to 0% at 80% or 90% removal. These results further indicate that current LLMs

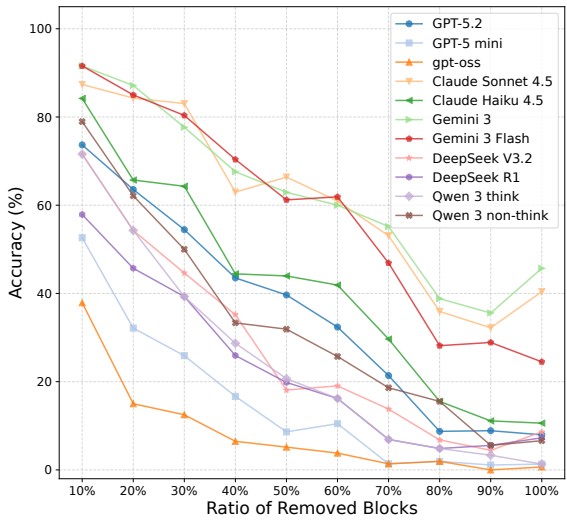

*(a)* Syntactic Accuracy

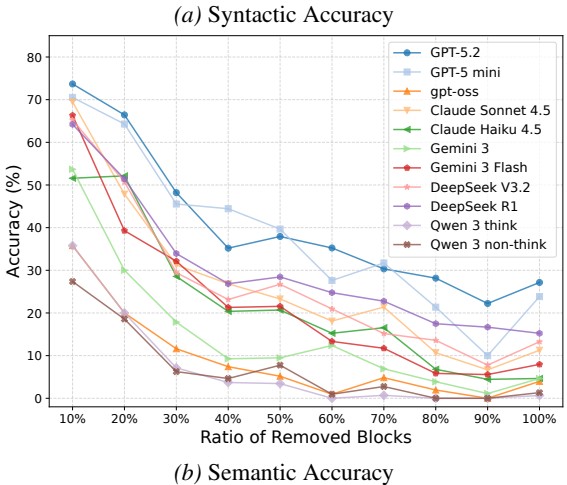

*(b)* Semantic Accuracy

*Figure 10.* Impact of semantic block removal on (a) Syntactic Accuracy, and (b) Semantic Accuracy.

have limited ability to reconstruct VCoT under substantial context removal.

***F2: Different decline trends for syntactic and semantic accuracy.*** Syntactic and semantic accuracies exhibit different decline patterns. Syntactic accuracy decreases smoothly as proofs are removed, whereas semantic accuracy drops sharply when removal increases from 10% to 40%, then declines more gradually from 40% to 100%.

This difference stems from their distinct roles. Syntactically, semantic proof blocks are largely independent, so each contributes similarly to overall correctness, producing a smooth decline as blocks are removed. Semantically, however, early removal disproportionately eliminates critical structural anchors in the verification logic; once these anchors fall below a threshold, logical continuity collapses, and models can no longer infer the proof structure, making the task uniformly difficult regardless of further removal.

*Table 4.* Syntactic and Semantic Accuracy across Proof Types. The **highest** and second-highest scores are highlighted.

| Model Name | Invariant | | Assertion | | Lemma | |
|---|---|---|---|---|---|---|
| | SynAcc | SemAcc | SynAcc | SemAcc | SynAcc | SemAcc |
| GPT-5.2 | 59.18 | **62.59** | 13.70 | 56.16 | 61.64 | **59.59** |
| GPT-5 mini | 12.24 | 39.46 | 2.74 | **63.01** | 23.97 | 54.79 |
| gpt-oss | 0.00 | 35.37 | 4.79 | 39.73 | 34.25 | 30.82 |
| Claude Sonnet 4.5 | 97.96 | 59.18 | **53.42** | 39.73 | 90.41 | 39.73 |
| Claude Haiku 4.5 | 89.80 | 40.82 | 8.22 | 47.26 | 69.18 | 36.99 |
| Gemini 3.1 Pro | **98.64** | 50.34 | 50.68 | 16.44 | **92.47** | 26.71 |
| Gemini 3 Flash | 97.96 | 44.90 | 28.08 | 38.36 | 88.36 | 26.71 |
| DeepSeek V3.2 | 89.80 | 37.41 | 4.11 | 56.16 | 55.48 | 46.58 |
| DeepSeek R1 | 79.59 | 37.41 | 7.53 | 52.74 | 34.93 | 49.32 |
| Qwen 3 think | 30.61 | 4.08 | 18.49 | 12.33 | 39.73 | 17.81 |
| Qwen 3 non-think | 73.47 | 12.24 | 28.08 | 11.64 | 47.26 | 16.44 |

*Table 5.* Syntactic and Semantic Accuracy across Proof Locations. The **highest** and second-highest scores are highlighted.

| Model Name | Front | | Middle | | End | |
|---|---|---|---|---|---|---|
| | SynAcc | SemAcc | SynAcc | SemAcc | SynAcc | SemAcc |
| GPT-5.2 | 63.85 | 63.85 | 35.38 | 54.62 | 66.15 | 73.85 |
| GPT-5 mini | 34.62 | **67.69** | 23.08 | **56.92** | 41.54 | **83.08** |
| gpt-oss | 10.00 | 30.77 | 6.92 | 15.38 | 22.31 | 38.46 |
| Claude Sonnet 4.5 | **93.08** | 62.31 | 63.08 | 46.92 | 80.77 | 64.62 |
| Claude Haiku 4.5 | 80.77 | 54.62 | 40.00 | 41.54 | 66.15 | 62.31 |
| Gemini 3.1 Pro | 90.77 | 54.62 | 64.62 | 28.46 | 80.00 | 36.92 |
| Gemini 3 Flash | 92.31 | 60.00 | **66.15** | 46.15 | **86.92** | 56.15 |
| DeepSeek V3.2 | 80.00 | 53.08 | 23.08 | 47.69 | 60.77 | 65.38 |
| DeepSeek R1 | 65.38 | 56.15 | 18.46 | 42.31 | 51.54 | 53.85 |
| Qwen 3 think | 49.23 | 15.38 | 33.08 | 10.77 | 63.85 | 28.46 |
| Qwen 3 non-think | 76.92 | 20.77 | 38.46 | 11.54 | 61.54 | 31.54 |

***F3: Syntactic-semantic imbalance.*** A syntactic-semantic imbalance is observed here. While Gemini 3.1 Pro exhibits low semantic accuracy but achieves relatively high syntactic accuracy, GPT-5.2 shows the opposite trend, with lower syntactic accuracy yet excelling in semantic reconstruction. This contrast highlights the pressing need for models to bridge the gap between grammatical fluency and verification reasoning to ensure more holistic reasoning.

### C.2. Performance across Proof Types

Using VCoT-Bench-Type, we evaluate LLM performance in both syntax and semantics across proof types, and report three key findings based on results in Table 4.

***F1: Assertions are the hardest for syntactic accuracy.*** Syntactic accuracy for loop invariants and lemma functions is consistently much higher than for assertions across most models. For example, DeepSeek V3.2 achieves 89.80% syntactic accuracy on loop invariants but drops sharply to 4.11% on assertions. This large gap reflects LLMs' limited familiarity with assertion syntax. Loop invariants impose looser grammatical constraints and closely resemble Rust code, making them easier to generate. Assertions, by contrast, involve substantially more complex grammar, particularly

*Table 6.* Accuracy of eight judge models across outputs from the ten evaluated models.

| Evaluated Models | OpenAI | | Claude | | Gemini | | DeepSeek | |
|---|---|---|---|---|---|---|---|---|
| *Accuracy of All Judges* | 5 mini | 5.2 | Sonnet 4.5 | Haiku 4.5 | 3 Pro | 3 Flash | R1 | V3.2 |
| GPT-5.2 | 94% | 78% | 72% | 54% | 94% | 86% | 78% | 75% |
| GPT-5 mini | 86% | 71% | 89% | 74% | 86% | 88% | 80% | 79% |
| gpt-oss 20b | 92% | 92% | 94% | 91% | 96% | 97% | 94% | 90% |
| Claude Haiku 4.5 | 92% | 77% | 86% | 80% | 91% | 92% | 83% | 86% |
| Claude Sonnet 4.5 | 91% | 74% | 87% | 73% | 87% | 88% | 76% | 74% |
| Gemini 3.1 Pro | 95% | 96% | 93% | 95% | 100% | 89% | 93% | 97% |
| Gemini 3 Flash | 94% | 82% | 80% | 80% | 94% | 80% | 82% | 81% |
| DeepSeek R1 | 90% | 76% | 88% | 77% | 86% | 86% | 76% | 77% |
| DeepSeek V3.2 | 92% | 79% | 92% | 76% | 90% | 89% | 82% | 79% |
| Qwen 3 non-think | 95% | 94% | 97% | 96% | 97% | 97% | 98% | 94% |
| Qwen 3 think | 99% | 94% | 97% | 94% | 97% | 100% | 95% | 95% |
| **Average** | **92.7%** | 83.0% | 88.6% | 80.9% | 92.5% | 90.2% | 85.2% | 84.3% |

*Table 7.* Self-evaluation bias of each judge model. Positive Self-Bias indicates higher false-positive rate on outputs from the same model family.

| *Self-Evaluation Bias* | | | |
|---|---|---|---|
| **Judge Model** | **FPR (Own)** | **FPR (Other)** | **Self-Bias** |
| GPT-5 mini | 6.7% | 7.5% | −0.8% |
| GPT-5.2 | 0.0% | 0.0% | 0.0% |
| Claude Sonnet 4.5 | 18.0% | 8.4% | +9.6% |
| Claude Haiku 4.5 | 5.0% | 2.2% | +2.8% |
| Gemini 3.1 Pro | 3.5% | 10.3% | −6.8% |
| Gemini 3 Flash | 17.5% | 9.8% | +7.7% |
| DeepSeek R1 | 4.5% | 2.6% | +1.9% |
| DeepSeek V3.2 | 0.5% | 1.8% | −1.3% |

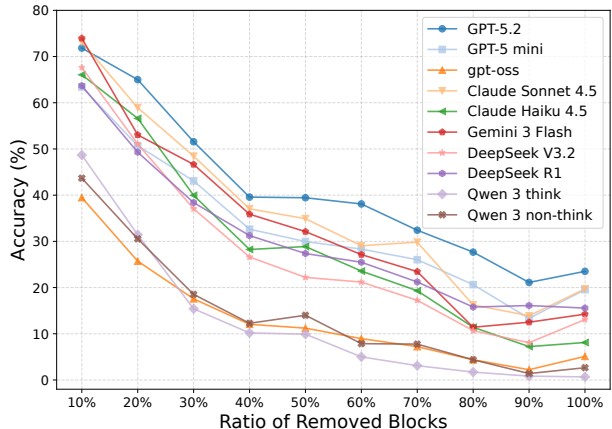

*Figure 11.* Overall accuracy under Best-of-10 sampling across varying proof-removal ratios.

strict type discipline that often requires explicit conversions to `int`, which models frequently miss. Lemma functions also involve specialized constructs such as `requires` and `ensures`, but demand less grammatical precision than assertions, resulting in syntactic accuracy between that of loop invariants and assertions.

***F2: Semantic accuracy varies across proof types.*** Assertions are not uniformly the most challenging proof type in terms of semantic accuracy, as model performance varies across proof types. Loop invariants require holistic reasoning across iterations, where GPT-5.2 and Claude Sonnet 4.5 perform best, achieving semantic accuracies of 62.59% and 59.18%, respectively. Assertions demand strong step-by-step inductive reasoning, on which GPT-5 mini performs best, outperforming GPT-5.2 (54.62%), Claude Sonnet 4.5 (46.92%), and Gemini 3.1 Pro (28.46%). Lemma functions require contextual reasoning over the main proof body and auxiliary properties, a setting in which the GPT family achieves the strongest overall performance.

***F3: Syntactic-semantic imbalance remains.*** A clear syntactic–semantic imbalance persists for Gemini 3.1 Pro and GPT-5.2. Gemini 3.1 Pro shows a striking disparity on loop invariants, achieving near-perfect syntactic accuracy (98.64%) but much lower semantic accuracy (50.34%). In

contrast, GPT-5.2 exhibits the opposite pattern, with lower syntactic accuracy (59.18%) but higher semantic accuracy (62.59%). This imbalance appears consistently across all proof types, indicating that the syntactic–semantic bias persists regardless of proof type.

### C.3. Performance across Proof Locations

Using VCoT-Bench-Loc, we evaluate LLM performance in both syntactic and semantic accuracy across proof locations and report three key findings based on the results in Table 5.

***F1: Middle Blocks are syntactically hardest.*** Across all models, syntactic accuracy on Front and End blocks consistently exceeds that on Middle blocks. For example, DeepSeek V3.2 achieves 90% syntactic accuracy on Front blocks and 60.77% on End blocks, but only 23.08% on Middle blocks. This gap stems from differences in syntactic structure: Front and End blocks typically contain short constraints with simple equalities or bounds, whereas Middle blocks usually require longer, more complex expressions with nested function calls and chained conditions to maintain state consistency, placing greater demands on grammatical understanding and leading to lower syntactic accuracy.

*Table 8.* Overall accuracy under Best-of-10 sampling across proof types and proof locations. The **highest** and second-highest accuracies are highlighted. Improvements over greedy decoding are shown in parentheses.

| Model Name | VCoT-Bench-Type | | | VCoT-Bench-Loc | | |
|---|---|---|---|---|---|---|
| *Results for overall accuracy* | | | | | | |
| | Invariant | Assertion | Lemma | Front | Middle | End |
| GPT-5.2 | 57.82 (+3.40) | 45.03 (+10.78) | **60.45 (+4.97)** | 62.88 (+4.61) | 48.27 (+7.12) | **70.00 (+4.42)** |
| GPT-5 mini | 31.63 (+8.16) | 43.21 (+10.33) | 43.66 (+7.36) | 54.04 (+5.00) | 43.08 (+6.16) | 62.88 (+2.88) |
| gpt-oss | 21.49 (+3.80) | 28.93 (+7.35) | 31.10 (+4.90) | 25.42 (+5.61) | 18.58 (+8.77) | 35.08 (+6.43) |
| Claude Sonnet 4.5 | **71.26 (+2.55)** | **46.92 (+7.37)** | 55.31 (+3.77) | **72.69 (+3.65)** | 49.62 (+3.27) | 68.46 (+2.50) |
| Claude Haiku 4.5 | 54.08 (+1.53) | 36.82 (+9.59) | 45.72 (+3.08) | 62.69 (+3.84) | 42.69 (+7.50) | 61.35 (+3.66) |
| Gemini 3 Flash | 59.86 (+2.21) | 36.47 (+7.87) | 43.15 (+1.54) | 70.38 (+3.26) | **52.50 (+4.81)** | 67.31 (+4.81) |
| DeepSeek V3.2 | 50.00 (+0.51) | 37.50 (+8.05) | 47.95 (+3.60) | 60.00 (+3.27) | 37.12 (+4.81) | 63.27 (+4.42) |
| DeepSeek R1 | 49.49 (+2.89) | 37.73 (+8.79) | 42.40 (+3.87) | 57.50 (+4.04) | 35.12 (+7.24) | 53.27 (+5.19) |
| Qwen 3 think | 12.07 (+2.04) | 16.44 (+4.97) | 25.00 (+2.57) | 27.31 (+4.43) | 18.85 (+4.04) | 40.15 (+4.57) |
| Qwen 3 non-think | 28.91 (+2.04) | 17.64 (+3.94) | 25.14 (+1.51) | 37.12 (+3.66) | 19.62 (+3.08) | 40.19 (+3.07) |

*F2: Middle blocks are semantically hardest* Across all models, semantic accuracy on Front and End blocks exceeds that on Middle blocks. Even for the best-performing model, GPT-5 mini, semantic accuracy reaches 67.69% on Front blocks and 83.08% on End blocks, but only 56.62% on Middle blocks. This pattern reinforces that current LLMs struggle significantly with connective reasoning in the middle of proofs.

*F3: Syntactic gaps exceed semantic gaps.* For syntactic accuracy, the performance gap across locations reaches 57.92%, while for semantic accuracy it narrows to 26.16%. This discrepancy suggests that current LLMs lack a robust understanding of Verus syntax but exhibit a relatively stronger grasp of verification semantics, which may build on code reasoning capabilities that most models are already trained on.

## D. Additional Evaluation Details

This section reports additional evaluation details that stress-test the main findings from two practical angles: whether the semantic judge is reliable and unbiased enough for evaluation, and whether multiple samples can mitigate VCoT-completion failures.

### D.1. Semantic Judge Validation

We further validate the reliability of the semantic judge using the same 100 stratified tasks described in Section 4.1. For each task, we collect generated outputs from the evaluated models and label each output by author consensus as semantically equivalent or non-equivalent to the ground-truth VCoT step. We then compare eight candidate judge models against these labels. This validation is designed to answer three questions: whether a judge agrees with author labels, whether it is overly permissive toward incorrect proofs, and whether it favors outputs from its own model family.

Table 6 reports agreement accuracy of each judge across outputs produced by the evaluated models. GPT-5 mini achieves the highest average accuracy (92.7%), followed closely by Gemini 3.1 Pro (92.5%), indicating that GPT-5 mini best matches author consensus under the primary agreement metric.

**Self-Evaluation Bias.** Accuracy alone does not capture whether a judge is overly permissive toward outputs from its own model family. We therefore examine false positive rate (FPR), which measures how often a judge incorrectly marks a non-equivalent proof as equivalent. Table 7 separates each judge's FPR into two groups: **FPR (Own)**, measured on outputs generated by models from the same family, and **FPR (Other)**, measured on outputs from different families. We compute **Self-Bias** as the difference between these two values, where a positive value indicates that the judge is more likely to falsely accept incorrect proofs from its own family.

The table shows that self-evaluation bias is not uniform across model families. GPT-5 mini has a slightly negative self-bias ($-0.8\%$), and GPT-5.2 has no detectable self-bias because its FPR is 0.0% on both own-family and other-family outputs. Gemini 3.1 Pro and DeepSeek V3.2 are also stricter on their own-family outputs than on other-family outputs. In contrast, Claude Sonnet 4.5 and Gemini 3 Flash show clear positive self-bias of 9.6% and 7.7%, respectively. Taken together, agreement and self-bias support GPT-5 mini as the default semantic judge: it has the highest agreement with author consensus and avoids positive same-family bias. These results support our choice to validate the semantic judge independently, rather than using a same-family judge as the default evaluator for each generator.

## D.2. Best-of-10 Sampling Results

We further evaluate whether repeated attempts can alleviate the fragility observed under greedy decoding. Figure 11 reports the overall accuracy of best-of-10 sampling across varying proof-removal ratios, showing how sampling changes performance as more VCoT context is removed. Table 8 reports the corresponding best-of-10 results for VCoT-Bench-Type and VCoT-Bench-Loc, with improvements measured against the greedy-decoding results in the main paper.

The main pattern is that sampling improves recovery probability but does not change the structure of the difficulty. In Figure 11, accuracy still decreases as more proof context is removed, which means that additional attempts cannot compensate for the loss of logical anchors in the VCoT. Table 8 shows the same conclusion from the Type and Location dimensions. The largest gains often appear on assertions and Middle blocks, where greedy decoding leaves more room for improvement, but their absolute accuracies remain low. Assertions remain challenging even with sampling: the best assertion accuracy reaches only 46.92% for Claude Sonnet 4.5. Similarly, Middle proof blocks remain difficult, with the best result reaching 52.50% for Gemini 3 Flash. These trends show that current LLMs benefit from multiple attempts but still lack robust deductive reasoning over VCoT structure.

