# OpenReview forum: "Can LLMs Reason Like Automated Theorem Provers for Rust Verification? VCoT-Bench: Evaluating via Verification Chain of Thought"
_ICML.cc/2026/Conference — ICML 2026 regular_

### Official Review · Reviewer_ZcKD · 2026-03-12

**Soundness:** 4
**Presentation:** 4
**Significance:** 4
**Originality:** 4
**Overall Recommendation:** 6
**Confidence:** 5

**Summary:**

The authors of this paper observe that previous research of program verification (especially verus-based research) treat the verification tool as a black box of answering fail/succeed, but actually verus can offer much more feedback information. This paper aims to leverage these information and feed them to LLM in a LLM-friendly way (the original text is hard to read). Specifically, this work opens the verification black box by introducing Verification Chain-of-Thought (VCoT), a novel concept that exposes the step-by-step deductive reasoning underlying formal verification. The authors present VCoT-Lift, an LLM-based framework that lifts low-level Z3 SMT solver proofs into high-level, human-readable Verus verification steps, providing transparent ground truth for evaluation. They also construct VCoT-Bench, a comprehensive benchmark of 1988 VCoT completion tasks that evaluates LLM understanding along three orthogonal dimensions: robustness to varying degrees of missing proofs, competence across different proof types (loop invariants, assertions, lemmas), and sensitivity to proof locations (front, middle, end). Through a systematic study of ten state-of-the-art LLMs, the paper reveals severe fragility in current models' reasoning capabilities, demonstrating that they fall well short of the deductive capabilities exhibited by automated theorem provers. The work makes important conceptual, technical, and empirical contributions to the field of LLM-assisted formal verification.

**Compliance With Llm Reviewing Policy:**

Affirmed.

**Final Justification:**

No more comment.

**Key Questions For Authors:**

I have no question to ask.

**Limitations:**

Yes. This paper does discuss limitations.

**Strengths And Weaknesses:**

This paper demonstrates exceptional quality across multiple dimensions. The research is exceptionally well-motivated: the authors clearly articulate why binary pass/fail evaluation is insufficient for assessing whether LLMs truly understand verification logic, and they convincingly argue that opening the verification black box is essential for meaningful progress in this domain. The conceptual contribution of VCoT is impactful, providing a principled framework for thinking about program verification. The technical design of VCoT-Lift is remarkably thorough and well-engineered: the four-stage pipeline (Transformer, Checker, Pruner, Repair) elegantly addresses the core challenges of soundness, completeness, and conciseness through carefully designed mechanisms including the Z3 rule hierarchy, abstraction filtering, and explicit mapping. The experimental evaluation is comprehensive, with 1988 tasks stratified across three orthogonal dimensions providing detailed measure of this method and also model capabilities. The empirical findings of this paper are well-supported by data.

Moreoever, the paper is well-written, with clear explanations, and easy to read.

---

> ### Author Rebuttal · Authors · 2026-03-31
>
> Thank you for the thoughtful and encouraging feedback. We are glad that the motivation, design of VCoT-Lift, and the benchmark evaluation are clear and well-supported, and we appreciate your recognition of the paper’s significance and clarity!

---

> > ### Author Rebuttal · Reviewer_ZcKD · 2026-04-01
> >
> > No more comment.

---

### Official Review · Reviewer_eCRb · 2026-03-12

**Soundness:** 3
**Presentation:** 2
**Significance:** 2
**Originality:** 3
**Overall Recommendation:** 3
**Confidence:** 4

**Summary:**

Current evaluations of LLMs on Rust formal verification (specifically using Verus) use global pass/fail metrics that fail to capture the model's ability to perform mathematical deduction. To address this, the authors introduce VCoT-Lift, a framework that translates verbose, machine-readable Z3 solver proofs into explicit, human-readable Verus verification steps. This translation is achieved via a multi-step pipeline consisting of a Transformer using a 3-tier Z3 rule hierarchy, a Checker module, a Proof Pruner, and a Proof Repair agent. All of these agents are instantiated by LLMs

Using data from VerusBench, the authors apply VCoT-Lift to construct VCoT-Bench, a diagnostic dataset of 1,988 "Verification Chain of Thought" completion tasks. The benchmark evaluates 10 models (including frontier models like GPT 5.2/Claude Sonnet 4.5 and open-source models) using a specialised LLM-based judge to decouple semantic understanding from strict Verus syntactic correctness. The evaluation measures performance across three dimensions: robustness to missing proofs, competence across proof types, and sensitivity to proof location.

**Compliance With Llm Reviewing Policy:**

Affirmed.

**Final Justification:**

I would like to thank the authors for the extensive rebuttal with additional experiments and explanations. This really helps me understand the impact and significance of the paper significantly better.

I agree that mining existing SMT solver proof-traces can be successfully used to train and improve AI models' ability to write formal code in Verus, a task which is clearly not yet saturated. However, I believe that the key limitations detailed in my original review remain. Reproducing the existing solver (e.g. Z3) performance with complex AI systems is an interesting goal, but relatively shallow and lacking in significance. While clearly out of scope for this work, I would like to see synthesis and evaluation of existing and/or trained models on problems outside the scope of current-generation SMT solvers, which would lead to significant progress in the research area. Therefore, I am inclined to keep my original score.

**Key Questions For Authors:**

**ATP Reasoning Alignment:** Given that usually there are multiple valid deductive paths to verify a program, why is strict alignment with Z3's internal logic the optimal evaluation criterion? Furthermore, how do you anticipate that improvements on this specific reconstruction task will translate to practical downstream improvements in real-world neuro-symbolic verification?

**Experimental Setup:** Could you provide a detailed breakdown of the experimental setup for the evaluated models, specifically including temperature settings, sampling algorithms, compute budgets, and context/output length restrictions?

**Missing Implementation Details:** Which specific LLMs were instantiated to power the VCoT-Lift pipeline, the Proof Repair agent, and the semantic judge? Will these be explicitly named in the camera-ready version to ensure reproducibility? Did you ablate the different set-ups? What were the costs of generating the benchmarks, ideally with a breakdown per sub-agent of the pipeline?

**Limitations:**

yes

**Strengths And Weaknesses:**

# Strengths
**Pipeline Design**: The four-stage architecture (Transformer, Checker, Pruner, Repair) is well developed and well-presented. The 3-tier Z3 rule hierarchy and the abstraction filter are clever ways to manage the massive context windows required for ATP traces and to restrict information to meaningful proof steps.

**Granular Evaluation Set-Up**: moving from a global pass/fail metric to a step-by-step verification trace evaluation by reconstructing Z3 traces is an original benchmark design. It allows for pinpointing exactly where models fail in multi-step reasoning chains and comparing multiple models across a significantly broader domain of tasks.

# Weaknesses

**ATP Reasoning Misalignment**: The underlying premise of the benchmark forces LLMs to reconstruct fine-grained Z3 reasoning. However, aligning LLM reasoning specifically with Z3 appears to be a somewhat arbitrary constraint.  In the proposed benchmark, an LLM might generate a perfectly sound, alternative form of reasoning that successfully verifies the program, but it would be penalised by the semantic judge simply because it deviates from Z3's internal logic. Consequently, it is entirely unclear why optimising a model to improve on this specific benchmark would translate to downstream improvements in general reasoning or practical verification capabilities. Lastly, in real-world deployments of the various Verus benchmarks quoted, LLMs and ATPs generally collaborate (LLMs provide high-level invariants while ATPs execute the granular "dirty work"), making strict Z3 emulation a questionable end-goal.

**Reproducibility:** The authors do not state which LLMs were instantiated to power the VCoT-Lift pipeline, the Proof Repair, or the semantic judge. There is no discussion of how the choice to instantiate the pipeline with certain models might influence the performance of the models in the evaluation. Additionally, the compute budgets, temperature settings, sampling algorithm and context and output length restrictions are not given in the paper.

**Self-Evaluation Bias:** Evaluating a model on a benchmark generated by itself (or a closely related frontier model), and using an LLM judge from the same family, introduces a significant risk of self-bias. The judge may artificially inflate scores for models that share their inherent syntactic or semantic style, skewing the comparative analysis against other (e.g., open-source) models.

**Figure Clarity:** Figures 2, 3, and 4 are overly dense, hard to read, and attempt to convey too much information simultaneously. Figure 3, in particular, desperately needs to be split apart or simplified.

---

> ### Author Rebuttal · Authors · 2026-03-31
>
> Thank you for your insightful feedback! Below is our response.
>
> **eCRb-Question1: ATP Reasoning Alignment**
>
> First, VCoT is not a replay of Z3’s low-level reasoning. VCoT-Lift explicitly bridges the gap between verbose solver traces and human-level reasoning by abstracting away low-level details, producing high-level, human-readable Verus steps (invariants, assertions, lemmas). The benchmark therefore evaluates reconstruction of a high-level deductive chain, not imitation of solver traces.
>
> Second, the evaluation task for LLMs is proof completion, not proof generation from scratch. While multiple valid proofs may exist, VCoT-Bench asks models to fill missing semantic blocks within a given, sound proof context, inferring the step that connects surrounding steps. Alternative proofs that do not align with this context fail because they do not complete the chain. Moreover, the evaluation emphasizes semantics: any logically correct step is accepted, even if expressed differently.
>
> Third, this directly impacts downstream neuro-symbolic verification. Generating proof hints for complex, state-heavy programs requires strong connective reasoning, which binary pass/fail metrics cannot capture. VCoT-Bench isolates this capability. For example, performance drops on middle proof steps reveal weaknesses in state tracking and multi-step composition. These skills are essential for synthesizing nontrivial invariants and assertions. By making them explicit and measurable, VCoT-Bench enables training LLMs that are more reliable for real-world verification.
>
> **eCRb-Question2: Experimental Setup**
>
> Temperature settings: We set temperature = 0 for all models to ensure deterministic, reproducible results. The only exception is GPT-5 mini, which does not expose a temperature parameter; we use its default API settings.
>
> Sampling algorithm: Our main experiments use greedy decoding for consistency and fair comparison. Following Reviewer uuC6’s suggestion, we also evaluate Best-of-N (N=10). This yields consistent but small improvements (+0.00% to +10.78%, typically +2% to +5%) and does not change any conclusions (see response to uuC6-Weakness2.2 for details).
>
> Compute budgets: A full cost breakdown across all models is available at https://anonymous.4open.science/r/VCoT-rebuttal-B8F5/cost.png.
>
> Context length: We impose no input limit; all tasks fit within every model’s context window.
>
> Output length: We set a maximum of 32,768 tokens for all models. No outputs reached this limit.
>
> **eCRb-Question3: Implementation Details**
>
> Semantic judge: We use GPT-5 mini. In details, we randomly sampled 100 stratified tasks from VCoT-Bench and evaluated eight judge candidates, the top LLMs at submission time from four families (OpenAI, Claude, Gemini, DeepSeek; all with temperature = 0), across all 11 evaluated models. We measure accuracy against author consensus. Results are shown at https://anonymous.4open.science/r/VCoT-rebuttal-B8F5/judge_accuracy.png. GPT-5 mini achieves the highest overall accuracy and is selected as the final judge.
>
> VCoT-Lift Pipeline: we use GPT-5.2 for the Proof Transformer, Checker, and Pruner, and Gemini 3 Pro for Proof Repair. Model selection is based on an ablation over 20 randomly sampled Verus-Bench problems, comparing GPT-5.2, Gemini 3 Pro, and Claude Sonnet 4.5 at each stage. We evaluate two aspects. For Z3-to-Verus reasoning (Transformer/Checker/Pruner), we measure the number of proof lines lifted (more lines increase the chance of recovering a complete Verus proof): GPT-5.2 achieves 45 lines on average, versus 21 (Gemini 3 Pro) and 7 (Claude Sonnet 4.5). For Verus syntactic/semantic refinement (Repair), we measure average repair rounds: Gemini 3 Pro performs best (4 rounds), followed by Claude Sonnet 4.5 (8 rounds) and GPT-5.2 (13 rounds).
>
> Cost: Constructing VCoT-Bench-Org cost $126.66 in total, with 91% spent on the transformer–checker loop, 1.9% on the pruner, and 7.1% on the repair stage.
>
> **eCRb-Weakness3: Self-Bias in LLM judge**
>
> Thank you for the suggestion. We added a targeted analysis to quantify potential self-bias in LLM judges. We measure self-evaluation bias via False Positive Rate (FPR): among all non-semantic-equivalent cases, the fraction labeled as equivalent (higher FPR = more over-acceptance). To isolate family effects, we compute: (1) FPR(Own): average FPR on outputs from the judge’s own family, and (2) FPR(Other): average FPR on outputs from other families. We define Self-Bias = FPR(Own) − FPR(Other), where positive values indicate over-acceptance toward its own family; negative values indicate stricter judgment; zero indicates no bias.
>
> The results of 8 candidate judges are shown at https://anonymous.4open.science/r/VCoT-rebuttal-B8F5/self_bias.png: Claude exhibits the strongest positive bias; GPT-family judges consistently show zero or negative bias. Our chosen judge, GPT-5 mini, has a Self-Bias of −0.8%, indicating no preference for its own family and slightly stricter evaluation.

---

> > ### Author Rebuttal · Reviewer_eCRb · 2026-04-03
> >
> > I would like to thank the authors for the extensive rebuttal with additional experiments and explanations. This really helps me understand the impact and significance of the paper significantly better.
> >
> > 1. ATP Reasoning Alignment
> > I found this response quite weak. Filling in the remaining proof steps, or generating a new proof from scratch, still has the exact same problem that multiple successful proof strategies might exist. I do not understand the comment "any logically correct step is accepted, even if expressed differently", when Verus is explicitly used to verify the reasoning traces generated. Could the authors clarify why matching Z3's proof-patterns with LLM-reasoning is a scalable and generalizable direction for reasoning systems, when the alternative of LLMs using solvers such as neuro-symbolic tools, such as Z3 solver itself, already exists and have shown great results.
> >
> > 2. OK
> > 3 & 4. This analysis is very interesting and useful. The paper would be significantly be improved with it's addition.

---

> > > ### Author Response · Authors · 2026-04-03
> > >
> > > We will definitely incorporate a detailed discussion based on our responses to points 3 and 4 raised by the reviewer. We believe this will significantly strengthen the paper. We sincerely thank the reviewer for the constructive feedback!
> > >
> > > We also greatly thank the reviewer for this thoughtful follow-up. We truly appreciate the opportunity to clarify points that we could not fully address in the first-round rebuttal because of space constraints.
> > >
> > > **Why learning the verification process matters even with neuro-symbolic tools**
> > >
> > > We agree with the reviewer that existing neuro-symbolic approaches have shown promising early results in generating proof hints. However, current state-of-the-art methods [1-4] still struggle on complex real-world Rust verification. Even with carefully designed LLM agents, the best reported performance is only 67% on complex system-level Rust programs [3], and fine-tuned open-source LLMs achieve only 49% on real-world Rust programs [4].
> > >
> > > Our central idea is that, to generate ***better proof hints*** for difficult programs, LLMs must learn the verification process itself from successful solver executions, step by step, rather than merely matching proof-hint patterns. We fully agree that the solver performs the actual verification. However, it has a key limitation: it does not retain experience across programs; each new program requires a fresh search, no matter how many similar problems it has solved before. LLMs, in contrast, can memorize, learn, and generalize from past successful verification trajectories. This creates a key opportunity: instead of treating proof hints as isolated syntactic patterns, LLMs can learn why an invariant is needed, how an assertion links two program states, and what dependency a lemma resolves. A model that internalizes these reasoning patterns can transfer them to new programs, generating proof hints that are not only syntactically plausible but also logically grounded in the verification structure.
> > >
> > > VCoT-Bench exists to measure and ultimately close this gap: it makes the step-by-step reasoning that separates shallow pattern matching from genuine verification understanding explicit, evaluable, and trainable.
> > >
> > > **Why multiple proofs do not invalidate VCoT-Bench**
> > >
> > > We agree that multiple proofs may exist for the same program. However, that is not what VCoT-Bench is designed to evaluate. Our goal is to test whether a model can understand a given verification process well enough to reconstruct its missing steps. The model is provided with a specific partially observed verification trajectory, and it must read and understand the surrounding steps in order to fill in the missing ones coherently.
> > >
> > > To evaluate the filled steps, we assess two aspects. First, we check syntactic correctness using Verus. As described in Section 4.1, Verus is used only in syntax-only mode to validate grammar; it is not used to judge logical correctness in our evaluation. Second, we assess semantic correctness using an LLM judge, which determines whether the reconstructed step is semantically equivalent to the ground-truth step (this is what we mean when we say that any logically correct step is accepted, even if it is expressed differently).
> > >
> > > Looking ahead, we plan to apply and further expand VCoT-Bench for fine-tuning LLMs to propose better proof hints. While multiple proofs may exist for the same program, learning from a specific successful verification process is still valuable because it teaches the model how valid verification reasoning unfolds step by step. This is similar to how a student can improve by studying one correct solution to a math problem. Moreover, as the training data grows, many different successful solutions across related programs will accumulate, allowing the model to gradually acquire broader and more transferable verification capabilities.
> > >
> > > **Closing Thanks**
> > >
> > > We are truly grateful for the reviewer’s thoughtful feedback, and we will definitely add a more detailed discussion of this concern to the paper. We believe this will further strengthen the paper!
> > >
> > > **References**
> > >
> > > [1] Chen, Tianyu, Shuai Lu, Shan Lu, Yeyun Gong, Chenyuan Yang, Xuheng Li, Md Rakib Hossain Misu et al. "Automated proof generation for rust code via self-evolution." ICLR (2025).
> > >
> > > [2] Yang, Chenyuan, Xuheng Li, Md Rakib Hossain Misu, Jianan Yao, Weidong Cui, Yeyun Gong, Chris Hawblitzel et al. "Autoverus: Automated proof generation for rust code." Proceedings of the ACM on Programming Languages 9, no. OOPSLA2 (2025): 3454-3482.
> > >
> > > [3] Yang, Chenyuan, Natalie Neamtu, Chris Hawblitzel, Jacob R. Lorch, and Shan Lu. "VeruSAGE: A Study of Agent-Based Verification for Rust Systems." arXiv preprint arXiv:2512.18436 (2025).
> > >
> > > [4] Di, Nongyu, Tianyu Chen, Shan Lu, Shuai Lu, Yeyun Gong, Peng Cheng, Jacob R. Lorch, Yuan Yao, and Xiaoxing Ma. "Reducing the Costs of Proof Synthesis on Rust Systems by Scaling Up a Seed Training Set." arXiv preprint arXiv:2602.04910 (2026).

---

### Official Review · Reviewer_Ag2E · 2026-03-13

**Soundness:** 2
**Presentation:** 3
**Significance:** 3
**Originality:** 3
**Overall Recommendation:** 4
**Confidence:** 3

**Summary:**

The paper introduces VCoT, a lifted version of Z3's internal solver reasoning. They propose exposing models to it. They also propose a benchmark that, using VCoT, tests the models' understanding of the verification process.

**Compliance With Llm Reviewing Policy:**

Affirmed.

**Final Justification:**

I recommend accepting the paper, given the changes the authors have promised to make. This is an exciting paper with a good premise, many good ideas and what appears to be a strong execution. Its main weakness remains the evaluation, but the authors' rebuttal has addressed my main concerns on that. Their reframing was useful.

**Key Questions For Authors:**

- Figure 5 could use some more explanation. There seem to be a lot of spikes -- what are they?
- It's a very interesting finding that reasoning seems to make models a bit worse at VCoT-Bench (F4 on page 8). It would be useful to have a look at the CoT generated to see what is happening. Did you investigate this further?
- What are VCoT's future applications? Is it possible that it might lead to improvements in verification?

**Limitations:**

Yes

**Strengths And Weaknesses:**

*Strengths*
- The writing is mostly clear.
- The question of how LLMs interact with verifiers is an interesting one and merits further study. The paper clearly has a novel contribution to this field of research.
- Modifying reasoning traces to be less unwieldy and easier to understand (both for models and people) is a good idea. VCoT-Lift has the potential to be an interesting technical contribution.

*Weaknesses*
- While I like the idea of VCoT-Bench, the authors' claim that it helps us ascertain whether models truly "understand" proofs needs more justification. Maybe "understand" is just not a good property to test for -- it feels a bit more like something a philosopher of mind may determine (cf. Chinese Room). This needs to be reworded.
- I also don't see any evidence that performance on VCoT-Bench directly correlates with verification performance (beyond the obvious fact that it does correlate with factors like model size which almost certainly also correlate with verification performance).
- Because of the above, it's not clear to me what VCoT-Bench is really measuring, and it doesn't seem to be an effective evaluation method.

- It's a very interesting finding that reasoning seems to make models a bit worse at VCoT-Bench (F4 on page 8). It would be useful to have a look at the CoT generated to see what is happening.
- Figure 5 could use some more explanation. There seem to be a lot of spikes -- what are they?
- The name VCoT is misleading. Chain of thought is human language that models are fine-tuned to produce, while VCoT is a reasoning trace (and the paper involved no fine tuning or model training).

I want to like this paper, but I find VCoT-Bench to be poorly motivated and ultimately not convincing as an evaluation method. That being said, I agree that the rest of this work is useful and could lead to interesting future directions. I strongly encourage to replace VCoT-Bench with some kind of SFT/RL on VCoT and an evaluation on how it changes the model's verification abilities for a resubmission.

---

> ### Author Rebuttal · Authors · 2026-03-31
>
> **Ag2E-Question1: Figure 5 spikes**
>
> The spikes reflect variability in verification complexity across programs. Verus-Bench includes both simple programs (e.g., single loops) and complex ones (e.g., nested loops, recursive lemmas). When lifted by VCoT-Lift, complex programs require many more intermediate proof steps, producing taller spikes.
>
> **Ag2E-Question2: CoT reasoning hurts performance**
>
> We analyzed CoT traces from Qwen-3-think on failed cases and identified two recurring failure patterns:
>
> 1. CoT often starts with a correct high-level strategy but gradually deviates. For example, in loop invariant reconstruction, the model identifies key sequence properties but, through extended reasoning, introduces unnecessary conditions or subtly alters predicates, breaking semantic equivalence. Longer chains increase this drift.
>
> 2. The model generates plausible but invalid Verus elements (e.g., nonexistent lemmas or tactics). These hallucinations propagate through the chain, leading to syntactic or semantic errors. In contrast, the non-thinking variant stays closer to contextual patterns.
>
> These behaviors reveal a mismatch between CoT-style reasoning and formal verification. While CoT benefits tasks that require decomposition, formal verification demands strict alignment with the verifier’s logical structure. Additional CoT reasoning steps do not aid decomposition; instead, they introduce noise and increase deviation, exposing the limitations of CoT reasoning in formal verification.
>
> **Ag2E-Question3: VCoT's future applications**
>
> First, VCoT can serve as a fine-grained training signal. Current ML-based verification systems rely mainly on binary supervision, whether the program verifies or not, which is weak and uninformative. In contrast, VCoT-Bench-Org exposes substantially richer reasoning supervision: on average, it contains 6.5× more proof lines, 13.4× more assertions, and 1.94× more lemma functions. These step-by-step traces can be used for fine-tuning, much like chain-of-thought supervision, to teach models why each proof step is needed rather than merely rewarding final success.
>
> Second, VCoT can guide proof generation. Our benchmark pinpoints where current models fail, especially on connective multi-step reasoning and state-dependent assertions. This diagnostic granularity enables more targeted proof-generation pipelines. For example, an agent could decompose verification into VCoT phases (as shown in Figure 8), generate and validate each phase incrementally, and use VCoT-style completion as a self-check before invoking the verifier. Such a structured generate-then-verify workflow is likely to be more robust than current end-to-end approaches that generate all proof hints at once and hope the verifier accepts.
>
> Third, the VCoT-Lift pipeline for lifting solver traces into source-level reasoning can be adapted to other SMT-backed verification systems, such as Dafny, Boogie, and F* with Z3, enabling VCoT construction at scale. This opens a path toward large, high-quality datasets of verification reasoning, which could support the next generation of verification-capable models.
>
> **Ag2E-Weaknesses2&3: What VCoT-Bench measures and its relevance to verification**
>
> Our experiments yield four actionable insights.
>
> (1) The sharp performance collapse as proof context is removed reveals that models reconstruct proofs by pattern-matching against nearby context rather than building internal representations of verification logic. This reframes the training challenge: models should be trained on degraded proof contexts at varying completion levels to incentivize genuine deductive reconstruction.
>
> (2) Thinking-mode variants can underperform simpler counterparts, challenging the assumption that CoT reasoning uniformly benefits logical tasks. Formal verification demands precision where verbose CoT reasoning introduces noise, suggesting that reasoning paradigms effective for natural language may need fundamental adaptation for formal domains.
>
> (3) The large performance gap across proof types reveals fundamentally different reasoning demands. This decomposition provides both a diagnostic lens for model weaknesses and a principled basis for targeted training and benchmark design for specific proof types.
>
> (4) Connective, intermediate reasoning is the central bottleneck. Models handle setup and closing steps well but struggle with middle steps requiring state propagation and multi-step composition, motivating methods that explicitly support intermediate-state tracking.
>
> **Ag2E-Weakness 6: name VCoT misleading**
>
> We use VCoT to highlight that verification reasoning is inherently sequential and compositional, where each step builds on prior ones. The connection to CoT is conceptual: both make implicit reasoning explicit through step-by-step traces. Importantly, while VCoT-Bench is designed for evaluation, VCoT traces can also serve as structured supervision for training, analogous to how CoT is used in NLP.

---

> > ### Author Rebuttal · Reviewer_Ag2E · 2026-04-02
> >
> > I thank the authors for their detailed comments.
> >
> > I agree that VCoT is useful, but my main concern is the comparatively weak benchmark, VCoT-Bench. Given my point from the review ["Maybe "understand" is just not a good property to test for -- it feels a bit more like something a philosopher of mind may determine (cf. Chinese Room). This needs to be reworded."] and your four actionable insights (which I agree are exciting), could you try to rephrase what VCoT-Bench's results say?

---

> > > ### Author Response · Authors · 2026-04-02
> > >
> > > We sincerely thank the reviewer for this thoughtful follow-up. We truly appreciate the opportunity to clarify points that we could not fully address in the first-round rebuttal due to space constraints.
> > >
> > > **Reframing “understand”**
> > >
> > > We agree that the term “understanding” carries philosophical implications that we do not intend to claim. We will revise the paper to adopt a precise, operational definition: VCoT-Bench evaluates whether a model can systematically reconstruct the explicit, step-by-step deductive reasoning chain underlying verification at the source-code level.
> > >
> > > Binary verification success can be achieved through superficial means, such as generating syntactically plausible proof hints that satisfy the verifier without reflecting the actual logical steps. In contrast, VCoT-Bench requires models to explicitly reconstruct missing deductive steps that are both syntactically valid and semantically faithful to the verifier’s reasoning trace. Success across different removal ratios, proof types, and proof positions therefore reflects a concrete capability: the ability to faithfully reproduce each step required to establish formal verification.
> > >
> > > **What VCoT-Bench’s results actually say**
> > >
> > > We fully agree with the reviewer that applying SFT or RL to VCoT is an exciting direction, and it is precisely our planned next step. However, effective training must be preceded by rigorous diagnosis. Before investing in SFT or RL, two fundamental questions must be answered: (1) Do current LLMs actually have systematic limitations in reconstructing verification reasoning? and (2) if so, what is the exact nature of those limitations? VCoT-Bench answers both.
> > >
> > > First, VCoT-Bench establishes that current LLMs have severe limitations in reconstructing proof steps. No current LLM can reliably reconstruct step-by-step deductive chains. Even the strongest model, Claude Sonnet 4.5, achieves only 71.58% accuracy with just 10% of proof blocks removed, and collapses to 17.22% when all blocks are removed. The other ten state-of-the-art models perform even worse. Without this systematic evaluation, the community might reasonably assume that existing models already possess adequate verification reasoning and only need scaling. Our results show this is not the case.
> > >
> > > Second, VCoT-Bench reveals that these limitations are not random but highly structured, directly informing how to design effective training. Prior to our study, it was entirely plausible that LLM failures in proof synthesis were uniform: random errors distributed uniformly across proof types, positions, and difficulty levels, in which case simply scaling up generic SFT on verification data would suffice. VCoT-Bench shows that this assumption is incorrect. The three sub-benchmarks act as complementary diagnostic lenses, each exposing a distinct failure mode and pointing to a different training strategy that contradicts naive SFT/RL assumptions:
> > >
> > > 1. VCoT-Bench-Ratio reveals *how* models reason. The sharp drop in performance up to 40% context removal, followed by a plateau, shows that models depend on a critical amount of local scaffolding rather than constructing deductive chains. This directly motivates a curriculum learning strategy: training should progressively remove proof context, from low to high removal ratios, so that models are forced to internalize verification logic instead of matching nearby proof patterns. Without this finding, a naive SFT strategy would likely train on complete proofs, reinforcing the same dependency on scaffolding.
> > >
> > > 2. VCoT-Bench-Type reveals *what* models struggle to reason about. Assertions are much harder than loop invariants or lemma functions; for example, Claude Sonnet 4.5 achieves 39.55% on assertions versus 68.71% on invariants. This indicates that the main bottleneck lies in precise, state-dependent reasoning rather than structural pattern recognition. That finding points to a targeted data strategy: SFT should oversample assertion-heavy proofs, and RL should use type-specific reward shaping instead of treating all proof types equally.
> > >
> > > 3. VCoT-Bench-Loc reveals *where* models fail within a proof.  Models’ consistent weakness on middle-position connective steps, which require state propagation and multi-step composition, compared with stronger performance on front and end steps, exposes a specific deficiency in intermediate reasoning. This suggests a corresponding training objective: RL rewards should explicitly value correct intermediate steps, not just final verification success, and training data should emphasize reasoning that connects different phases of a proof.
> > >
> > > In summary, VCoT-Bench provides a diagnostic foundation for training. It demonstrates the need for improved training by showing that current LLMs cannot reliably reconstruct verification reasoning, and it provides concrete guidance on how such training should be designed. We will include a detailed discussion in the next paper version.

---

### Official Review · Reviewer_uuC6 · 2026-03-15

**Soundness:** 2
**Presentation:** 2
**Significance:** 2
**Originality:** 3
**Overall Recommendation:** 2
**Confidence:** 3

**Summary:**

The paper concerns formal verification of Rust code with large language models. It argues that LLMs should understand (or be able to perform) the reasoning operations that the SMT solver performs in program verification. The paper develops a method for transforming a SMT solver trace into higher-level verification steps. It creates benchmark tasks based on predicting missing blocks of the higher-level steps, and evaluating with syntax check and LLM-as-a-judge.

**Compliance With Llm Reviewing Policy:**

Affirmed.

**Final Justification:**

My main concerns still stand, so I will retain my initial score of a 2.

**Key Questions For Authors:**

Please respond to the points in the review above; I do not currently have additional questions.

**Limitations:**

Yes.

**Strengths And Weaknesses:**

Strengths
- The paper presents a new idea related to mapping traces at the SMT solver level into a higher level format. While I am unclear about the benefit as a benchmarking task (see comments below), this may prove to be a useful idea to consider for training models in the future.

- Understanding potential limitations of language model reasoning in areas such as program verification is an interesting direction.

Weaknesses
- Unclear motivation: The paper motivates their benchmark by arguing that LLMs should understand the logic that the SMT solver is using.
I'm unclear on why an AI system (LLM, agent, etc) needs to understand or perform the same operations as the SMT solver, since the AI agent is complementary to the SMT solver. The agent can interface with the solver in the same way that a human can, and can write lemmas, invariants, etc. to get the verifier to pass. If the AI agent is able to do this, why would it additionally need to perform the same reasoning that the SMT solver does?

- The evaluation is based on LLM-as-a-judge. The motivation for the benchmark hinges on the assumption that LLMs do not understand the kinds of reasoning in the benchmark, so it's unclear why this evaluation metric is a good one. Furthermore, a natural baseline would be generating multiple candidates and selecting the best one according to the LLM judge, since the judge does not depend on priviledged information. However, I believe that experiments were done with a single generation per test example.

- The scope of the benchmark is narrow: it is specific to particular aspects of verified programming in Verus. When a conclusion is drawn from the benchmark results, it is therefore unclear how generalizable the conclusion is. More broadly, it's unclear what the broader significance or likely impacts on model development or understanding might be.

- Target audience: the paper content is largely details related to formal methods aspects rather than machine learning aspects. In my opinion the paper is better suited for a conference other than ICML.

- I wasn't sure what the actionable insights were from the experiments and findings (e.g., F1-F5 in 4.3 and F1-F2 in 4.4).  I didn't get a good insight for what influences the observed performance. The authors studied model scale a bit, but their conclusions were based on comparing models from different families (e.g., DeepSeek V3.2 and gpt-oss), so the training data and training procedure is a confounding factor.

---

> ### Author Rebuttal · Authors · 2026-03-31
>
> **uuC6-Weakness1: unclear motivation**
>
> We do not argue that LLMs should replicate SMT-level reasoning. VCoT-Lift explicitly abstracts away low-level Z3 proof traces, which consist of thousands of lines of machine-oriented symbolic steps, into high-level, human-readable Verus reasoning. Our goal is not to have LLMs operate at the SMT level, but to evaluate whether they understand verification logic at the source-code level, which is also how human developers reason.
>
> Our core argument is that generating correct proof hints requires understanding the underlying verification logic. Writing lemmas, invariants, or assertions is not purely syntactic; it depends on knowing what needs to be proved and why. For example, a valid loop invariant must hold at entry, be preserved across iterations, and ensure correctness at termination. These correspond exactly to the proof obligations enforced by the verifier. Without understanding these logical requirements, a model can only rely on superficial pattern matching.
>
> VCoT-Bench is designed to distinguish this difference. A model that truly understands the reasoning can reconstruct missing proof steps under partial context, while a pattern-based model will fail once surface cues are removed. This is precisely what we observe: performance drops sharply as context is reduced, and models struggle most on multi-step, state-dependent reasoning.
>
> This distinction is practically important. Existing systems (e.g., AutoVerus, SAFE, AlphaVerus) evaluate only whether the verifier accepts the output. This allows models to succeed via pattern matching without genuine understanding, which may not generalize to more complex, real-world verification tasks. VCoT-Bench provides fine-grained diagnostics to reveal where and why models fail, which is critical for improving them.
>
> **uuC6-Weakness2.1: LLM-as-judge validity**
>
> VCoT-Bench evaluates a generative task: constructing missing verification steps under uncertainty. In contrast, the judge performs a strictly easier discriminative task: assessing semantic equivalence between a candidate and a reference proof. This asymmetry is well established. Kirchner et al. [1] note that “discrimination is easier than generation,” and prior work shows LLM-based verifiers can reliably judge correctness even when generators fail [2-5]. To ensure reliability, we validate the judge against human consensus following standard practice [6–10]. Our LLM judge achieves 94% agreement, exceeding the 90% typically reported in code evaluation settings [9].
>
> **uuC6-Weakness2.2: missing Best-of-N baseline**
>
> Thank you for the suggestion. We added a Best-of-N baseline (N=10, temperature=0.8) to encourage diverse generations. Gemini 3 Pro is excluded as it was deprecated after submission. Results across
> removal ratios are at
> https://anonymous.4open.science/r/VCoT-rebuttal-B8F5/ratio_acc_sample.png, and across
> proof types and locations are at
> https://anonymous.4open.science/r/VCoT-rebuttal-B8F5/best-of-n.png. Best-of-10 yields consistent but small improvements across models and settings (+0.00% to +10.78%, typically +2% to +5%), without changing any claims in our paper.
>
> **uuC6-Weakness3.1: generalizability beyond Verus**
>
> While VCoT-Bench is instantiated in Verus, the reasoning it evaluates is not Verus-specific. Its core proof constructs are fundamental to deductive verification and appear across systems such as Dafny, F*, Coq, and Lean. Syntax differs across frameworks, but the underlying logical structure is shared. Our benchmark is designed around this structure. Its three dimensions (Ratio, Type, and Location) operate on logical dependencies rather than Verus-specific features. Accordingly, our findings are structural: performance degradation under context removal, difficulty with connective middle-step reasoning, and challenges with state-dependent assertions reflect intrinsic properties of multi-step deductive reasoning, not artifacts of Verus.
>
> **uuC6-Weakness3.2: broader impact**
>
> Please refer to Ag2E-Question3.
>
> **uuC6-Weakness4: unsuitable for ICML**
>
> We respectfully disagree. First, we address a core ML question: whether LLMs perform genuine deductive reasoning or rely on pattern matching. Second, VCoT-Lift is an LLM-centric pipeline. Third, our study is ML-focused, evaluating ten LLMs and analyzing key phenomena across three dimensions. Finally, VCoT-Bench is a reusable ML resource, providing structured supervision for training and diagnosis toward stronger models for verification.
>
> **uuC6-Weakness5: actionable insights unclear**
>
> Please refer to Ag2E-Weaknesses2&3.
>
>
> **References**
>
> [1] https://arxiv.org/abs/2407.13692
>
> [2] https://arxiv.org/abs/2110.14168
>
> [3] https://arxiv.org/abs/2305.20050
>
> [4] https://arxiv.org/abs/2304.14317
>
> [5] https://arxiv.org/abs/2410.02184
>
> [6] https://arxiv.org/abs/2406.04770
>
> [7] https://arxiv.org/abs/2406.11939
>
> [8] https://arxiv.org/abs/2306.04751
>
> [9] https://arxiv.org/abs/2509.01494
>
> [10] https://arxiv.org/abs/2406.18403

---

> > ### Author Rebuttal · Reviewer_uuC6 · 2026-04-01
> >
> > Thank you for the response. I have read through it and appreciate the responses. However my main concerns about the significance/utility of the benchmark (which were also voiced by two other reviewers) and the use of LLM as judge still remain. There are no further followups needed, the authors replied with detailed responses. I will maintain my score.

---

> > > ### Author Response · Authors · 2026-04-07
> > >
> > > We sincerely thank the reviewer for the continued engagement. Although no further follow-up was requested, we would like to briefly clarify the two core concerns that we could not fully address in the first-round rebuttal due to space constraints.
> > >
> > > **Significance of the VCoT-Bench**
> > >
> > > We agree that existing AI systems have shown promising early results in generating proof hints to assist solvers for program verification. However, current state-of-the-art methods still fall far short on complex real-world Rust programs; the best reported success rate is only 67% on system-level Rust verification [1].
> > >
> > > Our central idea is that generating correct proof hints for complex programs requires LLMs to internalize the verification process itself, not merely match proof-hint patterns. We fully agree that the solver is the component that performs the actual verification. However, it has a key limitation: it retains no experience across programs. Each new program requires a fresh search, no matter how many similar problems it has solved before. LLMs, in contrast, can learn and generalize from past successful verification trajectories. This creates a concrete opportunity: instead of treating proof hints as isolated syntactic patterns, LLMs can learn why an invariant is needed, how an assertion connects two program states, and what dependency a lemma resolves. A model that internalizes these entire reasoning patterns can transfer them to new programs, generating much better proof hints that are not only syntactically plausible but logically grounded in the verification structure. VCoT-Bench exists to measure and ultimately close this gap.
> > >
> > > **Utility of VCoT-Bench**
> > >
> > > We fully agree with the reviewer that applying VCoT for model training is an exciting direction, and it is precisely our planned next step. However, effective training must be preceded by rigorous diagnosis. Before investing in SFT or RL, two fundamental questions must be answered: (1) Do current LLMs actually have systematic limitations in reconstructing verification reasoning? and (2) if so, what is the exact nature of those limitations? VCoT-Bench answers both.
> > >
> > > First, VCoT-Bench establishes that current LLMs have severe limitations in reconstructing proof steps. No current LLM can reliably reconstruct step-by-step deductive chains. Without this systematic evaluation, the community might reasonably assume that existing models already possess adequate verification reasoning and only need scaling.
> > >
> > > Second, VCoT-Bench reveals that these limitations are not random but highly structured, directly informing how to design effective training. Prior to our study, it was entirely plausible that LLM failures in proof synthesis were uniform: random errors distributed uniformly across proof types, positions, and difficulty levels, in which case simply scaling up generic SFT on verification data would suffice. VCoT-Bench shows that this assumption is incorrect. The three sub-benchmarks act as complementary diagnostic lenses, each exposing a distinct failure mode and pointing to a different training strategy that contradicts naive SFT/RL assumptions:
> > >
> > > 1. VCoT-Bench-Ratio reveals *how* models reason. The sharp drop in performance up to 40% context removal, followed by a plateau, shows that models depend on a critical amount of local scaffolding rather than constructing deductive chains. This directly motivates a curriculum learning strategy: training should progressively remove proof context, from low to high removal ratios, so that models are forced to internalize verification logic instead of matching nearby proof patterns. Without this finding, a naive SFT strategy would likely train on complete proofs, reinforcing the same dependency on scaffolding.
> > >
> > > 2. VCoT-Bench-Type reveals *what* models struggle to reason about. Assertions are much harder than loop invariants or lemma functions; for example, Claude Sonnet 4.5 achieves 39.55% on assertions versus 68.71% on invariants. This indicates that the main bottleneck lies in precise, state-dependent reasoning rather than structural pattern recognition. That finding points to a targeted data strategy: SFT should oversample assertion-heavy proofs, and RL should use type-specific reward shaping instead of treating all proof types equally.
> > >
> > > 3. VCoT-Bench-Loc reveals *where* models fail within a proof. Models’ consistent weakness on middle-position connective steps, which require state propagation and multi-step composition, compared with stronger performance on front and end steps, exposes a specific deficiency in intermediate reasoning. This suggests a corresponding training objective: RL rewards should explicitly value correct intermediate steps, not just final verification success, and training data should emphasize reasoning that connects different phases of a proof.
> > >
> > >
> > > **References**
> > >
> > > [1] Yang et al. "VeruSAGE: A Study of Agent-Based Verification for Rust Systems." arXiv preprint arXiv:2512.18436 (2025).

---

### Decision · Program_Chairs · 2026-04-30

**Decision:**

Accept (regular)

**Comment:**

This paper introduces VCoT-Lift, a framework that transforms low-level Z3 SMT solver traces into high-level, human-readable Verus verification steps; and VCoT-Bench, a benchmark of VCoT completion tasks designed to evaluate LLMs' fine-grained reasoning capabilities in Rust program verification. The main weaknesses are less about technical invalidity and more about claim calibration and external relevance; for example, the paper sometimes overstates what VCoT-Bench measures by tying it to “understanding” or to the need for models to mirror solver reasoning, while the connection between benchmark performance and real end-to-end verification success is argued rather than demonstrated directly. The author's response was thoughtful and appears to have resolved a substantial fraction of the discussion; as a result, most of the concern is toward a narrower disagreement about how much value to assign to the benchmark and how strongly to interpret its results.

My reading is that the remaining issues are real but not fatal. The paper should not be read as proving that VCoT-Bench is the definitive measure of verification ability, but rather as introducing a useful diagnostic benchmark that opens the black box of LLM-based verification and reveals structured failure modes. On balance, I recommend weak acceptance, while encouraging the authors in the camera-ready version to further soften claims about what "understanding" means, emphasize more on the benchmark’s diagnostic role, and clearly acknowledge that transfer to broader proof generation remains an important limitation.